

# Impacts of fire severity and salvage-logging on soil carbon fluxes in a boreal forest

Julia Kelly[1], Stefan H. Doerr[2], Johan Ekroos[3,4], Theresa S. Ibáñez[5], Md. Rafikul Islam[1], Cristina Santín[6,7], Margarida Soares[1], Natascha Kljun[1]

[1]Centre for Environmental and Climate Science (CEC), Lund University, Lund, Sweden
[2]Geography Department, Swansea University, Swansea, UK
[3]Department of Agricultural Sciences, University of Helsinki, Helsinki, Finland
[4]Helsinki Institute of Sustainability Science (HELSUS), University of Helsinki, Finland
[5]Department of Wildlife, Fish and Environmental Studies, Swedish University of Agricultural Sciences, Umeå, Sweden
[6]Research Institute of Biodiversity (IMIB), Spanish National Research Council-University of Oviedo, Spain
[7]Biosciences Department, Swansea University, UK

*Correspondence to*: Julia Kelly (julia.kelly@cec.lu.se)

**Abstract.** The long-term carbon storage capacity of the boreal forest is under threat from the increasing frequency and intensity of wildfires. In addition to the direct carbon emissions from combustion during a fire, the burnt forest often turns into a net carbon emitter after fire, leading to large additional losses of carbon over several years. Understanding how quickly forests recover after a fire is therefore vital to predicting the effects of fire on the forest carbon balance. We present soil respiration and $CH_4$ fluxes, soil chemistry, site microclimate and vegetation survey data from the first four years after a wildfire in a *Pinus sylvestris* forest in Sweden. This is an understudied part of the boreal biome where forest management decisions interact with disturbances to affect forest growth. We analysed how fire severity and post-fire salvage-logging affected the soil carbon fluxes. The fire did not affect soil $CH_4$ uptake. However, soil $CO_2$ emissions were significantly affected by the presence or absence of living trees after the fire and post-fire forest management. Tree mortality due to the high-severity fire, or the salvage-logging of living trees after low-severity fire, led to immediate and significant decreases in soil respiration. Salvage-logging of dead trees after high-severity fire did not alter soil respiration compared to when the trees were left standing. However, it did significantly slow the regrowth of vegetation. Sites where trees had been left standing after the fire also had double the density of *Pinus sylvestris* seedlings from natural regeneration compared to sites where the trees had been salvage-logged. Our results highlight that the impact of salvage-logging on the soil carbon fluxes depends on the fire severity but that logging always slows the natural recovery of vegetation post-fire.



## 1 Introduction

Boreal forests store more carbon (C) than any other forest biome, but their C stores are at risk from increasingly frequent and severe wildfires (Bradshaw and Warkentin, 2015; Zhao et al., 2021). In 2018, an unprecedented number of forest fires broke out across Sweden due to prolonged drought, burning an area ten times larger than the annual mean (SOU, 2019). The increasing frequency of wildfires in these slow growing forests is reducing their capacity to accumulate and store C over the long-term, and is altering the vegetation communities that establish after a fire (Walker et al., 2019; Burrell et al., 2022; Mack et al., 2021).

During fire, large amounts of C can be released into the atmosphere. The burnt ecosystem may continue to be a net C emitter until newly established or surviving vegetation regrows sufficiently to turn the forest back into a C sink. These post-fire C losses can account for a significant proportion of the total C loss caused by forest fires (Ueyama et al., 2019). Forest floor respiration ($R_{ff}$, i.e. the sum of autotrophic respiration from forest floor vegetation and heterotrophic respiration from soil microbes) is the dominant component of post-fire ecosystem C emissions. In undisturbed Swedish boreal forests, $R_{ff}$ contributes 82% of total ecosystem respiration and can be the main driver of differences in the annual net C balance between forest stands (Chi et al., 2021). Changes in $R_{ff}$ become even more important in determining the net C balance of a stand after a fire because gross primary production (GPP) partly or completely stops immediately after a fire.

Soil respiration tends to decrease post-fire, especially after high-severity fires leading to high mortality of the trees and understory. This loss of vegetation not only reduces autotrophic respiration but also heterotrophic respiration since the root exudates that many microbes depend on are no longer produced. In addition, the associated high soil burn severity can kill soil microbes and combust a large proportion of the soil organic layer, further reducing heterotrophic respiration (Xu et al., 2022; Zhou et al., 2023). In boreal North America, high-severity stand-replacing fires are typical, and this type of fire has been the focus of most boreal forest fire research (O'neill et al., 2002; Amiro et al., 2003; Köster et al., 2017).

In contrast, less is known about the impact of low-severity surface fires that are typical across boreal Eurasia on forest carbon fluxes (Rogers et al., 2015). In boreal Eurasia, forests are dominated by tree species adapted to resist fire, such as larch and Scots pine (Rogers et al., 2015). During a low-severity fire in these forests, the understory vegetation and part of the soil organic layer is consumed but most (if not all) trees survive. In a Chinese boreal forest, Hu et al. (2017) found that low-severity fire only caused a significant reduction in autotrophic, but not in heterotrophic, respiration compared to unburnt plots. This apparently more complex response of soil respiration to low-severity fire needs further investigation to help quantify how these fires affect the forest carbon budget across boreal Eurasia.





Fire can also affect the emission and uptake of methane ($CH_4$) by soil bacteria and vegetation. Dry, oxic soils act as a small
$CH_4$ sink, consuming 5% of all global $CH_4$ emissions (Saunois et al., 2020). Measuring both soil respiration and methane
fluxes after forest fires and how they change with time since fire is therefore vital for understanding how fast the forest C
balance recovers post-fire. There are fewer studies assessing the impact of fire on forest $CH_4$ fluxes compared to $CO_2$ fluxes.
Previous studies reported contrasting effects: increases (Jaatinen et al., 2004; Burke et al., 1997), decreases (Kulmala et al.,
2014), and no significant effects (Köster et al., 2018) on soil $CH_4$ uptake after boreal forest fires. The processes controlling
methane uptake by forest soils are also not well understood. For example, the thickness of the soil humus layer, which is
affected by fire, has been shown to be both positively and negatively correlated with $CH_4$ uptake (Mcnamara et al., 2015; Saari
et al., 1998).

Over half of the global boreal forest is managed (Astrup et al., 2015), yet few studies have explicitly considered how forest
management after fire may affect the boreal forest C budget. Salvage-logging (cutting of burnt trees) is a common practice
after fire in managed boreal forests (Nappi et al., 2011; Skogsstyrelsen, 2023) and an additional disturbance that may amplify
the fire impacts (Leverkus et al., 2018). Between 1-21 years after wildfire in hemiboreal and boreal forests, neither Parro et al.
(2019) nor Kelly et al. (2021) found significant differences in soil respiration between salvage-logged and unlogged forests
after a stand-replacing fire. However, the impact of management after low-severity fire, the most common fire type in the
intensively managed northern European boreal forest, has not previously been considered.

Our study contributes to filling the above research gaps by analysing a time series of soil C flux measurements (soil respiration
and $CH_4$) collected during the first four years after a major forest fire in boreal Sweden. The extensive Ljusdal fire of 2018
enabled us examine sites affected by low and high-severity fire, and with or without post-fire salvage-logging, providing
unique insights into the impacts of both fire and management on post-fire forest recovery in an understudied part of the boreal
forest. This work builds on a previous study from the first post-fire year (Kelly et al., 2021). Here, we focus on two research
questions: i) what is the impact of fire severity on post-fire soil C fluxes and ii) what is the impact of salvage-logging compared
to leaving the trees standing, after both high and low-severity fire? We answer these questions with the help of several years
of data on soil C fluxes, soil nutrient content, soil microclimate and vegetation regrowth.
**2 Methods**
**2.1 Study area and design**
The study area is in central Sweden (61°56'N 15°28'E, 220 m a.s.l) and had a mean annual temperature of 3.8 °C and mean
annual precipitation of 652 mm during the study period 2019-2022 (SMHI, 2023; Ytterhogdal station 263 m a.s.l. and 40 km
northwest of the site). It sits in a wide, flat valley, dominated by managed *Pinus sylvestris* forests with smaller areas of *Picea*
*abies* and *Betula sp*. The understory vegetation consists of low shrubs (*Vaccinium vitis-idaea, Vaccinium myrtillus,*



*Arctostaphylos uva-ursi, Empetrum nigrum, Calluna vulgaris*) and bryophytes (*Pleurozium schreberi, Dicranum* sp.,
*Polytrichum juniperinum, Cladonia* sp., *Cetraria* sp.). The soils are Podzols. The Ljusdal wildfire was ignited by lightning in
July 2018 and burned 8995 ha, making it one of the largest Swedish forest fires of this and the last century (Drobyshev et al.,
2015; Sou, 2019). The burnt area included areas affected by high-severity fire, which we define as having complete tree
mortality, and areas affected by low-severity fire, where most of the soil organic layer and understory vegetation was
combusted, but all the trees survived. More details about the fire and study area can be found in Kelly et al. (2021).

After the fire, we established five sites in mature *Pinus sylvestris* forests that were affected by contrasting fire severity and
post-fire management treatments (salvage-logging versus unlogged, replanted versus natural regeneration; Figure 1). Forest
owners decided how their plots would be managed after the fire and we did not influence this decision, nor were we involved
in carrying out the chosen post-fire treatments. We present the results from these sites split into three groups:
'Fire severity' group: comparing an unburnt site (UM) with a low-severity fire (LM), high-severity fire (HM). These three
sites are all part of a nature reserve created after the fire. No salvage-logging occurred at LM or at HM and the sites have been
allowed to regenerate naturally.
'Salvage-logging after low-severity fire' group: comparing LM (unlogged) with a site that experienced low-severity fire but
was salvage-logged (SLM). Salvage-logging occurred within 10 months after the fire. In late spring 2019, soil scarification
was performed, creating ridges with the charred and organic soil layers and furrows of exposed mineral soil. Seeds of *Pinus*
*sylvestris* were spread after soil scarification.
'Salvage-logging after high-severity fire' group: comparing HM (unlogged) with a site that experienced high-severity fire and
was then salvage-logged (SHM). The SHM site was salvage-logged 6 months after the fire, but it was not scarified. Two year
old *Pinus* sylvestris seedlings were planted at SHM in 2020, two years after the fire.

We deliberately chose not to compare groups 2 and 3 since the salvage-logged sites in these two groups experienced different
post-fire management treatments (i.e. scarification or not, spreading of seeds versus planting seedlings). The characteristics of
all the sites are summarized in Table 1 and Figure 1. The sites are located within 3 km of each other, ensuring the same weather
conditions. Note that the first year of soil flux and chemistry data from sites UM, LM, HM and SHM are presented in Kelly et
al. (2021), sapflow and tree growth data from UM and LM in Dukat et al. (2024), eddy-covariance data from SLM in Kelly et
al. (2024) and soil fungal and bacterial growth and respiration data from UM, LM, HM and SHM in Soares, et al. (in review).







**Table 1. Description of the sites in the study area affected by the 2018 Ljusdal wildfire. Uncertainties are ± SE. DBH = diameter at breast height. The forest floor refers to the combined litter layer (including bryophytes if present), charred layer (at the burnt sites only) and soil organic layer.**

| Description | UM | LM | HM | SLM | SHM |
|---|---|---|---|---|---|
| Site name | Unburnt Mature | Low-severity Mature | High-severity Mature | Salvage-logged, Low-severity Mature | Salvage-logged, High-severity Mature |
| Fire severity | No fire | Low | High | Low | High |
| Post-fire management | None (nature reserve) | Standing living trees with charring of the lower trunk, natural regeneration (nature reserve) | Standing dead burnt trees, natural regeneration (nature reserve) | Living trees salvage-logged within 10 months after fire, soil scarification and spreading of *Pinus sylvestris* seeds in late spring 2019 (commercial plantation) | Dead trees salvage-logged 6 months after fire, no soil preparation, *Pinus sylvestris* seedlings planted in spring 2020 (commercial plantation) |
| Charred forest floor layer depth (mm)* | NA | 8 ± 1 | 10 ± 0 | 0, 11 ± 1 | 9 ± 1 |
| Total forest floor layer depth (mm)* | 149 ± 4 | 37 ± 2 | 25 ± 1 | 0, 26 ± 3 | 23 ± 2 |
| Tree age in 2018 | 60-70 | 70-90 | ~100 | 54 | 73 |
| Mineral soil type | Sand | Sand | Sand | Silt loam | Sand |

*At all sites, charred layer depth and/or total forest floor layer depth were measured in May 2019 except at SLM where they were measured in June 2020, see Section 2.4. At SLM, two measurements of charred and total forest floor layer depths are given to represent the furrows with exposed mineral soil (0 mm forest floor and charred layer) and the ridges (forest floor remaining).




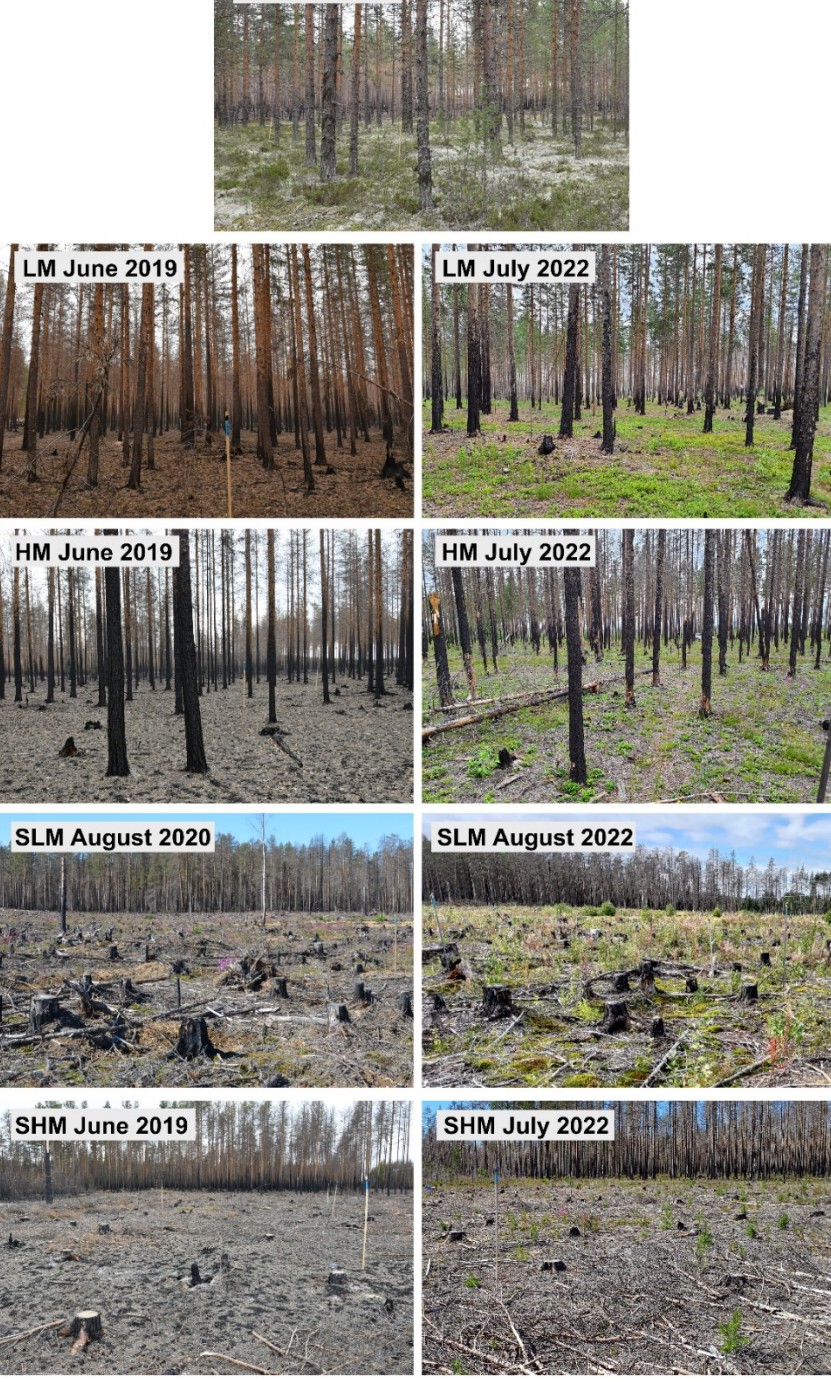

134

**Figure 1. Site photos from the first or second year after the fire and the fourth year after the fire. Site characteristics: UM (unburnt), LM (low-severity fire), HM (high-severity fire, dead trees left standing), living trees left standing), SLM (low-severity fire, living trees salvage-logged) and SHM (high-severity fire, dead trees salvage-logged).**



**2.2 Soil greenhouse gas flux measurements**

Between 2019 and 2021, we conducted manual soil $CO_2$ and $CH_4$ measurements with a dark chamber monthly between June-September at all sites except SLM (where measurements started in 2020). We refer to these dark chamber $CO_2$ measurements as forest floor respiration ($R_{ff}$) to highlight that they include respiration from the soil and from any understory vegetation growing in the collars. In 2022, we measured the fluxes only in July and August. The difference in the length of the sampling period had little effect on the soil greenhouse gas results: an analysis using only July-August data from all years (Figure S1 and Tables S1 and S2) showed the same trends in the $R_{ff}$ data and only minor differences in the $CH_4$ data as when the June-September data was included (Figure 3a, 3c and Tables 2 and 3).

The soil flux measurements were conducted using a static chamber on 10 collars per site in 2019 with an Ultra-portable Greenhouse Gas Analyser (Los Gatos Research Inc.) and 12 collars per site in 2020-2022 with an LI-7810 Gas Analyser (LI-COR Environmental). At all sites, 10 of the collars were arranged in two transects at 10 m intervals while the two additional collars were randomly placed within the site. To account for the soil scarification at SLM, five collars from the transects plus two random collars were located in the ridges with intact organic soil, while five collars were placed in the furrows of mineral soil. We combined the data from the SLM collars in our main statistical analysis. Plots of the fluxes separated by soil type are available in Figure S2.

The circular collars had a diameter of 16 cm and extended 10 cm into the soil. The dark chamber flux measurements followed the method and conversion from concentration to flux described in Kelly et al. (2021). These included using a 5 min chamber closure time, 150 second duration for the calculation of the linear regression of gas concentration versus time and selecting regressions with the highest $R^2$ where $p<0.001$ and NRMSE$<0.2$. After each flux measurement, the soil temperature at 5 cm depth was measured at two locations just outside the collar (thermometer HI98501 Hanna Instruments Ltd.) and the soil water content (SWC) integrated over 0-6 cm depth was measured at three locations (SM300 sensor in 2019, ML3 sensor in 2021-2022 with a HH2 moisture meter, Delta-T Devices Ltd.). These soil temperature and air pressure measurements from SLM (EC100 barometer, Campbell Scientific, Inc.) were used as inputs to the ideal gas law to transform the gas concentration data to fluxes. Negative $CO_2$ or $CH_4$ fluxes indicate an uptake by the ecosystem whereas positive fluxes indicate emission to the atmosphere.

**2.3 Soil greenhouse gas flux data analysis**

We fit linear mixed effects models to the soil flux data (one model per group and gas flux) to assess whether there were significant differences in $R_{ff}$ and $CH_4$ fluxes between the sites within each group. The groups were (see Section 2.1): fire severity (UM, LM and HM), salvage-logging after low-severity fire (LM and SLM) and salvage-logging after high-severity fire (HM and SHM). For the $R_{ff}$ and $CH_4$ fluxes, we modelled the data from every year between 2019-2022, using site and time since fire as fixed effects. The $R_{ff}$ models also had soil temperature at 5 cm depth ($T_{soil}$; from manual measurements during



the soil flux data collection) as a covariate whereas the $CH_4$ models had SWC as a covariate because there was a stronger
correlation between $CH_4$ and SWC than soil temperature and vice versa for $R_{ff}$. We did not include both soil temperature and
SWC in the same model to avoid issues of collinearity due to the strong correlation between these two factors. We included
soil temperature and SWC as covariates in the models since they are key drivers of the soil fluxes (Davidson and Janssens,
2006; Smith et al., 2000). It also enabled testing for significant differences in the fluxes between sites at a specific SWC or soil
temperature. All the fixed effects were centered at their mean value. Collar ID nested within site was included as a random
effect to account for the multiple measurements per collar. Interactions between soil temperature or SWC and site or time since
fire were only included in the models if significant. $R_{ff}$ data were log-transformed to ensure a normal distribution; this was not
necessary for the $CH_4$ data. We included a variance structure (VarIdent, described in Zuur et al., 2009) with site as the covariate
in the models to account for the different variances in model residuals between sites. The model residuals met assumptions of
equal variance and normal distribution.

ANOVAs, followed by Tukey's post-hoc tests, were conducted on the models to establish whether there were significant
differences in the fluxes between sites within each group and over time since the fire. All the mixed effects model analysis
was performed in R using the nlme package (Pinheiro et al., 2023). Model fit is described using marginal $R^2$ ($R^2_{marg}$, the
variance explained by the fixed effects), conditional $R^2$ ($R^2_{con}$, the variance explained by the fixed and random effects) and
root mean square error (RMSE) expressed in the units of the response variable. $R^2_{marg}$ and $R^2_{con}$ are calculated using the
performance package (Lüdecke et al., 2021) based on Nakagawa and Schielzeth (2013).

When presenting the $R_{ff}$ data, we show both $R_{ff}$ and $R_{ff}$ normalized ($R_{ff\_norm}$) to 15°C soil temperature and 10% SWC, to
eliminate the effects of variations in weather conditions during each sampling round. The 15°C value was chosen because it is
close to the mean soil temperature across all measurements (16°C) and has been used as a reference temperature previously
(e.g., Lasslop et al., 2010) whilst the 10% SWC is the mean SWC across all measurements. The $R_{ff}$ normalization was based
on a model from Carey et al. (2016), where $\log(R_{ff}) = a + b \times T_{soil} + c \times T_{soil}^2 + d \times SWC$. $T_{soil}$ and SWC are from the manual
measurements taken at the same time as the soil flux data, whilst a, b, c and d are fitted coefficients. Each site was modelled
separately. Model $R^2$ for the soil respiration models was between 0.26 and 0.47. We did not normalize the $CH_4$ fluxes because
the data was not well represented by any model.

## 2.4 Soil sampling and chemical analysis

Soils were sampled at all sites once per year at the start of the growing season (May or June) from 2019 to 2022. The entire
forest floor layer, which includes the charred organic layer (when present), the soil uncharred organic layer, and any litter,
mosses or lichens present was collected as a single sample within a 20 cm × 20 cm square every 2 m along two 20 m-long
transects within a few meters of the soil flux collars. In the center of the 20 cm x 20 cm square, a sample of the top 0-2 cm of
the mineral layer was also collected. We sampled at different locations every year. The 20 samples collected per site and layer





were pooled to create four composite samples. The forest floor and mineral soil composite samples were analysed for total
concentrations of carbon (C), nitrogen (N) and phosphorus (P); C:N ratio; water-soluble C and phosphorus (P); ammonium
($NH_4^+$); nitrate ($NO_3^-$); bioavailable P (Melich P); effective cation exchange capacity (ECEC); electrical conductivity (EC) and
pH. The protocols for the sample preparation and chemical analysis are described in Kelly et al. (2021). No carbonates were
present in the lithology of the study area, and no carbonates were formed by combustion during the fire, so all soil carbon is
assumed to be organic. Due to the small sample size per site and year, we did not perform any statistical tests on these data.
**2.5 Microclimate**
At UM, LM and SLM, soil temperature and soil moisture were monitored continuously during the whole study period with
Campbell Scientific CS655 sensors (6 at each site, installed at 7.5 and 15 cm depth). Soil temperature probes also provided a
shallow continuous measurement (3 cm, 7.5 cm, 15 cm depth, 107 Thermistors, Campbell Scientific, Inc.). At SLM, furrows
of exposed mineral soil and ridges of intact burnt organic layer were monitored separately. In addition, two TOMST TMS-4
loggers at all sites (one at each end of the soil flux collar transect) captured time series of soil temperature (7.5 cm depth),
near-surface air temperatures (1.5 cm and 14 cm above the soil surface) and soil water content (2-13.5 cm depth). At SLM,
four loggers were installed, two in the furrows and two in the ridges. The loggers were installed after soil thaw at the start of
the 2022 growing season. The manufacture-provided sun shields were used above the 1.5 and 15 cm air temperature sensors.
The loggers recorded data every 10 mins.

To convert the raw soil moisture data from the TMS4-loggers to SWC, we calibrated the sensors by fitting a linear regression
($R^2$ between 0.48-0.81) against the CS655 sensor data at UM, LM and SLM. The TMS-4 data from HM and SHM were
calibrated using the LM calibration curve because no CS655 sensors were installed at these sites.
**2.6 Vegetation recovery**
We surveyed the coverage of the understory vegetation at the burnt sites in July 2020-2022 (unburnt site only in 2022). Within
a 25 cm x 25 cm quadrat, the proportional cover of each vascular plant species and of all bryophytes was visually estimated
following Delin (2021). We surveyed 12 quadrats per site along two transects of the same length but a few meters away from
the soil flux collar transects. To estimate the total understory vegetation cover within the quadrat, we summed the cover from
all vascular species and bryophytes in each quadrat. We also counted the number of *Pinus sylvestris* seedlings within 4 round
plots (3 m radius) per site in 2022. We specifically targeted *Pinus sylvestris* since this was the dominant tree species at the
sites before the fire and since it is the main species used for commercial production in the region. At the SHM site it was
possible to determine which seedlings were from natural regeneration and which had been planted after the salvage-logging
based on the seedling height and position.



To test for significant differences in proportional vegetation cover between the sites and over time since the fire within each
site group, we modelled total vegetation cover (vascular plants and bryophytes, excluding *Pinus sylvestris* seedlings), as well
as vascular plants and bryophytes separately using beta regressions (R package betareg, Cribari-Neto and Zeileis, 2010). To
avoid values of 0 and 1 in beta regressions, we transformed proportional plant cover using the formula: plant cover proportion
$\times$ (n − 1) + 0.5] / n, where n = the number of survey plots in the compared site group (Smithson and Verkuilen, 2006). We
fitted one regression per plant and site group, using a log link function and site and year as dependent variables (their interaction
was not significant). Chi-square likelihood ratio tests were then used to test for significant differences between the sites and
years.

## 3 Results

### 3.1 Forest floor $CO_2$ fluxes

Fire severity had a significant impact on forest floor respiration ($R_{ff}$; Table 2 and Figure 2a). $R_{ff}$ was significantly lower at HM
(mean $\pm$ SE = 1.03 $\pm$ 0.04 $\mu$mol m$^{-2}$ s$^{-1}$) compared to both LM (2.23 $\pm$ 0.12 $\mu$mol m$^{-2}$ s$^{-1}$; Tukey test p<0.0001) and UM (2.53 $\pm$
0.10 $\mu$mol m$^{-2}$ s$^{-1}$; p<0.0001) during the whole study period. Significant differences in $R_{ff}$ at LM compared to UM only
appeared in the third and fourth years after fire (Tukey test p = 0.03 and <0.0001, respectively), when $R_{ff}$ was lower at LM,
but not as low as at HM (Figure 3a). As a result, there was a significant interaction between site and time since fire in the fire
severity model. $R_{ff}$ at UM was much higher in 2022 compared to previous years. The high $R_{ff}$ values at UM in 2022 were
driven by a few measurements of very high $R_{ff}$ in August 2022. We could not find any fault with the measurements and
therefore retained them in the analysis.
The salvage-logged, low-severity fire site (SLM) had consistently and significantly lower $R_{ff}$ (1.20 $\pm$ 0.06 $\mu$mol m$^{-2}$ s$^{-1}$)
compared to the low-severity fire site where the living trees had been left standing after the fire (LM; p = 0.004; Figures 2b
and 3b, Table 2). There was a significant interaction between site and time since fire because $R_{ff}$ at LM decreased over time
since the fire, whereas $R_{ff}$ increased slightly at SLM (Figure 3b; Table 2).
After high-severity fire, salvage-logging (SHM; 1.05 $\pm$ 0.04 $\mu$mol m$^{-2}$ s$^{-1}$) had no effect on the $R_{ff}$ compared to leaving the
dead trees standing (HM; Figure 2c; Table 2). Time since fire had a significant impact on $R_{ff}$ at both sites: $R_{ff}$ decreased during
the first three years post-fire after which it started increasing again (Table 2, Figure 3a).

### 3.2 Forest floor $CH_4$ fluxes

All sites were $CH_4$ sinks during the entire study period (Figure 3c, Table 3). The mean ($\pm\sigma$) $CH_4$ flux was -1.17 $\pm$ 0.04 nmol
m$^{-2}$ s$^{-1}$ at UM, -1.36 $\pm$ 0.03 nmol m$^{-2}$ s$^{-1}$ at LM and -1.16 $\pm$ 0.03 nmol m$^{-2}$ s$^{-1}$ at HM, -1.89 $\pm$ 0.12 nmol m$^{-2}$ s$^{-1}$ at SLM and -
1.17 $\pm$ 0.05 nmol m$^{-2}$ s$^{-1}$ at SHM. Neither fire severity nor salvage-logging after high severity fire had a significant effect on




the soil $CH_4$ fluxes (Figure 2d, f). However, after low severity fire soil $CH_4$ uptake was significantly higher after salvage-
logging (SLM) compared to leaving the trees standing (LM; Figure 2e). In the SLM/LM model, the differences in $CH_4$ flux
between the sites varied significantly over time. All our $CH_4$ models had much higher conditional $R^2$ (which includes random
and fixed effects) compared to marginal $R^2$ (only fixed effects). This highlights the large variability in the $CH_4$ uptake between
the collars at each site since collar ID was included as a random effect in the models (Table 3).

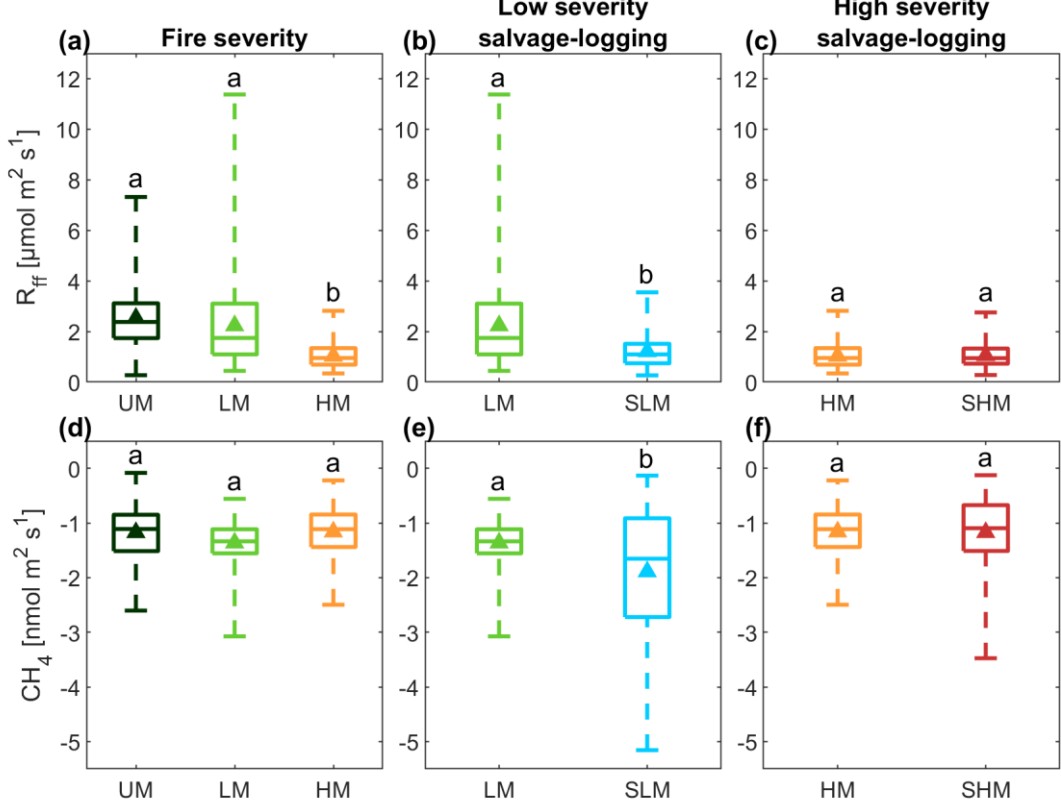

**Figure 2. Boxplots of all soil flux measurements from 2019 to 2022 (raw data), where different letters above the boxplots indicate**
**significant differences between the sites based on the ANOVA results in Table 2 and Tukey post-hoc tests. Triangles show the mean**
**flux. (a-c) forest floor respiration, (d-f) soil $CH_4$ flux. Data is divided into groups for fire severity (a, d), low-severity fire and salvage-**
**logging (b, e) and high-severity fire and salvage-logging (c, f). Site characteristics: UM (unburnt), LM (low-severity fire), HM (high-**
**severity fire, dead trees left standing), living trees left standing), SLM (low-severity fire, living trees salvage-logged) and SHM (high-**
**severity fire, dead trees salvage-logged).**



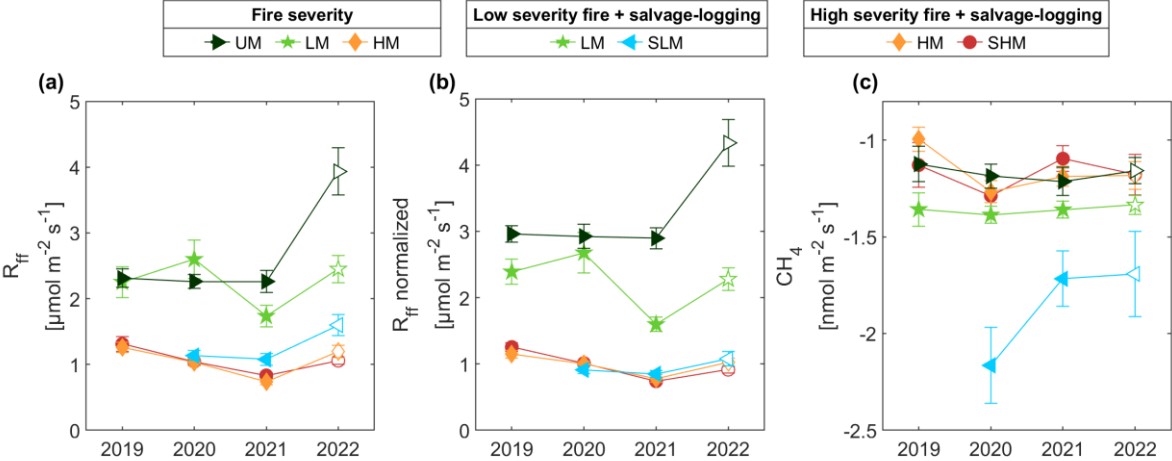

**Figure 3. Growing season means (±SE) of (a) forest floor respiration ($R_{ff}$; raw data), (b) normalized $R_{ff}$ ($R_{ff\_norm}$) and (c) soil $CH_4$ flux (raw data) for 2019-2022. $R_{ff\_norm}$ is normalized to soil temperature of 15 °C and 10% soil water content. In 2019-2021, the averages include June-September data (closed symbols), while in 2022 they include July-August data (open symbols). Site characteristics: UM (unburnt), LM (low-severity fire), HM (high-severity fire, dead trees left standing), living trees left standing), SLM (low-severity fire, living trees salvage-logged) and SHM (high-severity fire, dead trees salvage-logged).**





**Table 2. Results of ANOVAs on the mixed effects models of the forest floor respiration flux data, 2019-2022. Low + SL is low-severity fire followed by salvage-logging, High + SL is high-severity fire followed by salvage-logging. $T_{soil}$ is soil temperature at 5 cm depth. Interactions were only included in the models if significant. The site × $T_{soil}$ interaction was not significant in any model. df = numerator degrees of freedom, denominator degrees of freedom, $R^2_{marg}$ = marginal $R^2$ (variance explained by fixed effects), $R^2_{con}$ = conditional $R^2$ (variance explained by the fixed and random effects), RMSE = root mean square error (µmol $CO_2$ m$^{-2}$ s$^{-1}$). Statistically significant effects are marked in bold (p<0.05).**

| Group | Site | | | Tsoil | | | Time | | | Site × Time | | | Model fit | | |
|---|---|---|---|---|---|---|---|---|---|---|---|---|---|---|---|
| | df | F | p | df | F | p | df | F | p | df | F | p | $R^2_{marg}$ | $R^2_{con}$ | RMSE |
| Fire severity | 2, 33 | 41.42 | **<0.001** | 1, 438 | 320.25 | **<0.001** | 1, 438 | 4.84 | **0.029** | 2, 438 | 8.82 | **<0.001** | 0.49 | 0.67 | 0.34 |
| Low + SL | 1, 22 | 10.59 | **0.004** | 1, 213 | 115.27 | **<0.001** | 1, 213 | 2.77 | 0.098 | 1, 213 | 7.34 | **0.007** | 0.33 | 0.70 | 0.33 |
| High + SL | 1, 22 | 0.01 | 0.934 | 1, 294 | 324.29 | **<0.001** | 1, 294 | 35.31 | **<0.001** | - | - | - | 0.46 | 0.55 | 0.29 |

**Table 3. Results of ANOVAs on the mixed effects models of the soil $CH_4$ flux data, 2019-2022. Low + SL is low-severity fire followed by salvage-logging, High + SL is high-severity fire followed by salvage-logging. SWC is soil water content at 0-6 cm depth. Interactions were only included in the models if significant. df = numerator degrees of freedom, denominator degrees of freedom, $R^2_{marg}$ = marginal $R^2$ (variance explained by fixed effects), $R^2_{con}$ = conditional $R^2$ (variance explained by the fixed and random effects), RMSE = root mean square error (nmol $CH_4$ m$^{-2}$ s$^{-1}$). Statistically significant effects are marked in bold (p<0.05).**

| Group | Site | | | SWC | | | Time | | | Site × SWC | | | Site × Time | | | Model fit | | |
|---|---|---|---|---|---|---|---|---|---|---|---|---|---|---|---|---|---|---|
| | df | F | p | df | F | p | df | F | p | df | F | p | df | F | p | $R^2_{marg}$ | $R^2_{con}$ | RMSE |
| Fire severity | 2, 33 | 0.73 | 0.489 | 1, 440 | 139.63 | **<0.001** | 1, 440 | 0.38 | 0.540 | - | - | - | - | - | - | 0.15 | 0.56 | 0.26 |
| Low + SL | 1, 22 | 4.46 | **0.046** | 1, 212 | 19.93 | **<0.001** | 1, 212 | 4.34 | **0.038** | 1, 212 | 7.10 | **0.008** | 1, 212 | 7.01 | **0.009** | 0.18 | 0.57 | 0.43 |
| High + SL | 1, 22 | 0.07 | 0.789 | 1, 294 | 42.63 | **<0.001** | 1, 294 | 1.38 | 0.241 | - | - | - | - | - | - | 0.04 | 0.64 | 0.27 |



## 3.2 Forest floor and mineral soil layer chemistry

In the forest floor layer, none of the nutrients showed a marked trend over time since fire (Figure 4). Many nutrients (bioavailable P, ECEC, water-soluble C, $NH_4^+$, C%, N%) showed large interannual variability within a site. Water-soluble C, water-soluble P, and EC (Figures 4d, g, l) were notably higher at the unburnt site compared to all burnt sites. In addition, the low-severity fire site (LM) had higher water-soluble C, water-soluble P and EC compared to the high-severity burnt site (HM). For both salvage-logging groups (LM vs SLM and HM vs SHM), the salvage-logged site tended to have lower concentrations of water-soluble C and P and lower EC compared to the unlogged site. At SLM, the soil samples were only collected in the areas where the organic soil layer remained. But samples from the mineral soil in those areas suggest that in areas with exposed mineral soil due to soil scarification, the concentration of all the nutrients except $NO_3^-$ was much lower than in the organic layer (Figure S3f). The mineral soil layer had a similar chemical composition at all sites and over time after the fire (Figure S3).



**Figure 4. Time series of mean (± SE) soil nutrient content in the forest floor layer at all sites, see Figure S2 for the mineral layer results. Site characteristics: UM (unburnt), LM (low-severity fire), HM (high-severity fire, dead trees left standing), living trees left standing), SLM (low-severity fire, living trees salvage-logged) and SHM (high-severity fire, dead trees salvage-logged).**



**3.3 Microclimate**

All sites had almost identical air temperatures at 14 cm above the soil surface (Figure S4) with small differences appearing in mean daily air temperature at 1.5 cm (Figure S4) and the largest differences between the sites in the soil temperature at 7.5 cm depth (Figure 5a-d).

Within the fire severity group (UM, LM and HM), HM experienced the largest range of soil temperatures, with maximum temperature exceeding that of the LM and UM sites by 3°C (Figure 5a). In spring 2022, the soil thawed at least two weeks earlier at the two burnt sites (HM and LM) compared to the unburnt site (UM). During the peak growing season in July, the daily mean soil temperature was on average 2.3°C higher at HM than at UM, and 0.6°C higher at LM compared to UM. The high-severity fire site had consistently lower soil moisture (SWC) than the low-severity and unburnt sites although the difference was small (mean SWC at HM 5.4% compared to 6.8% at LM and 6.7% at UM; Figure 5e).

In both salvage-logging groups (LM vs SLM and HM vs SHM), the salvage-logged site experienced a larger range of soil temperatures than the unlogged site (Figures 5d). The difference was especially pronounced at SLM, where the maximum reached 25.9°C compared to 17.5°C at LM. At SHM, 22.6°C was reached compared to 20.4°C at HM. Throughout the growing season, daily mean soil temperatures were higher at the salvage-logged than at the unlogged sites in both groups (Figure 5b, c). SLM had much higher mean SWC than LM (22.2% compared to 6.8%; Figure 5f, h). In the high-severity group, however, the salvage-logged site had similar mean SWC as the unlogged site (SHM mean = 3.1%, HM mean = 5.4%; Figure 5g, h).

**Figure 5.** (a-c) Daily mean soil temperature at 7.5 cm depth for each site group (d) boxplot of all soil temperature measurements for all sites, (e-g) daily mean soil water content (SWC; when soil was not frozen) integrated over 2-13.5cm depth, (h) boxplot of all SWC measurements for all sites. In the boxplots, the triangle shows the mean. Site characteristics: UM (unburnt), LM (low-severity fire), HM (high-severity fire, dead trees left standing), living trees left standing), SLM (low-severity fire, living trees salvage-logged) and SHM (high-severity fire, dead trees salvage-logged).



**3.4 Vegetation recovery**


Within the fire severity group (LM vs HM), LM had significantly higher total and vascular vegetation cover than
HM (Figures 6a, b). LM was the burnt site with the highest total vegetation cover in 2022 (26%). For the salvage-
logging after low-severity fire group (LM vs SLM), there was no significant difference in total cover between the
two sites because SLM had significantly lower vascular cover, but also significantly higher bryophyte cover than
LM (Figures 6a, b, c). After high-severity fire and salvage-logging, the SHM site had the lowest total vegetation
cover of all the burnt sites (8% in 2022). SHM had significantly lower total and vascular vegetation cover compared
to HM (Figure 6a). Total vegetation cover increased significantly between 2020 and 2022 for all 3 groups, but
only the fire severity group showed a significant increase in vascular cover over time and none of the groups had
significant changes in bryophyte cover over time (Table 4, Figure S5).

In terms of *Pinus sylvestris* seedling density, SLM had the lowest density (1415 seedlings ha$^{-1}$, pine seeds spread
after salvage-logging) followed by SHM (3625 seedlings ha$^{-1}$, of which 1150 ha$^{-1}$ were planted and 2476 ha$^{-1}$ were
from natural regeneration), HM (4156 seedlings ha$^{-1}$ natural regeneration) and LM with the highest density (6189
seedlings ha$^{-1}$ natural regeneration).









**Table 4. Results of Chi-squared tests on the beta regressions of the total, vascular and bryophyte understory vegetation cover. Low + SL is low-severity fire followed by salvage-logging, High + SL is high-severity fire followed by salvage-logging. Statistically significant effects are marked in bold (p<0.05).**

| Group | Site | | Time | | Model Fit |
|---|---|---|---|---|---|
| | Chi-sq | p | Chi-sq | p | Pseudo $R^2$ |
| *Total vegetation cover* | | | | | |
| Fire severity | 7.06 | **0.008** | 11.03 | **0.004** | 0.30 |
| Low + SL | 1.78 | 0.182 | 7.58 | **0.023** | 0.17 |
| High + SL | 7.30 | **0.007** | 6.30 | **0.043** | 0.26 |
| *Vascular plants* | | | | | |
| Fire severity | 6.23 | **0.013** | 6.13 | **0.047** | 0.22 |
| Low + SL | 18.85 | **<0.001** | 2.13 | 0.345 | 0.37 |
| High + SL | 5.10 | **0.024** | 4.36 | 0.113 | 0.21 |
| *Bryophyte cover* | | | | | |
| Fire severity | 0.13 | 0.723 | 3.01 | 0.222 | 0.11 |
| Low + SL | 6.68 | **0.010** | 5.67 | 0.059 | 0.28 |
| High + SL | 3.08 | 0.079 | 1.81 | 0.405 | 0.14 |

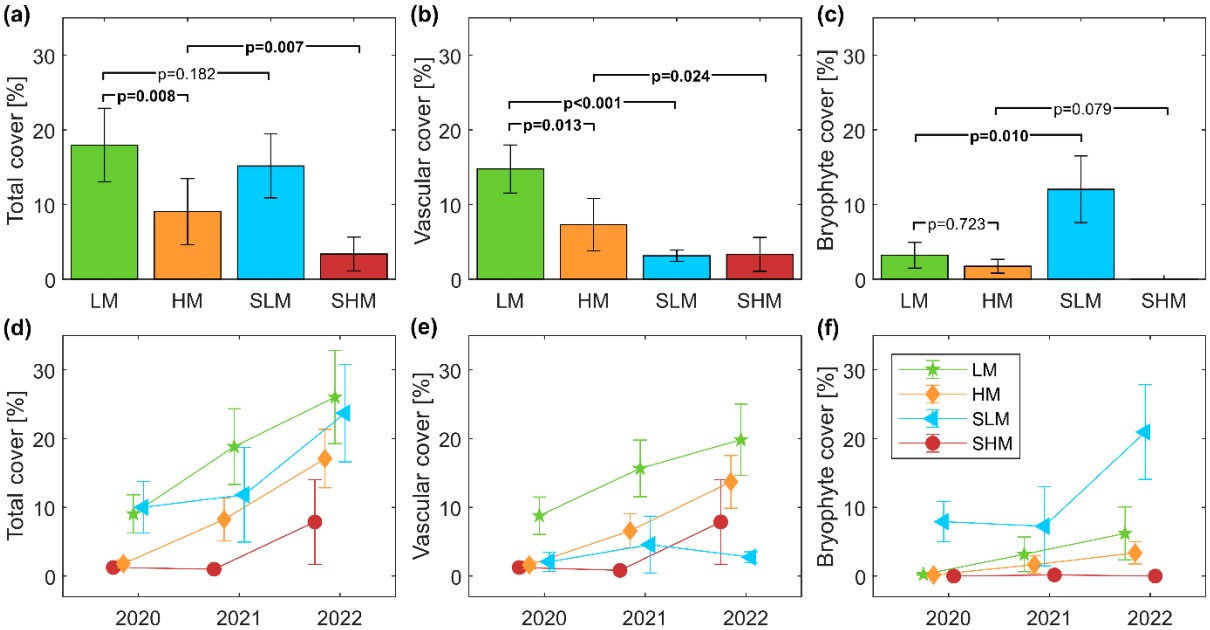

Figure 6. Mean ± SE of raw data from 2020-2022 of (a) total vegetation cover, (b) vascular cover and (c) bryophyte cover in the understory within all the burnt sites. SHM mean and SE of bryophyte cover were <1% and are not visible on the plot. P values show results of chi-square tests for significant differences between sites within each group (LM vs HM, LM vs SLM and HM vs SHM; Table 4). (d-f) annual mean ± SE of raw total vegetation, vascular or bryophyte cover, respectively, at the burnt sites. Site characteristics: LM (low-severity fire), HM (high-severity fire, dead trees left standing), living trees left standing), SLM (low-severity fire, living trees salvage-logged) and SHM (high-severity fire, dead trees salvage-logged).



## 4 Discussion

### 4.1 Effects of fire severity on forest floor CO$_2$ fluxes

Forest floor respiration (R$_{ff}$) was significantly lower for the first four years after high-severity fire (HM) compared to low-severity fire (LM) or no fire (UM; Figures 2 and 3). Similarly, Parro et al. (2019) and Ludwig et al. (2018) found that soil respiration 1-21 years after high-severity fire was only 50% or less of that measured in unburnt Eurasian boreal forest stands. In their meta-analysis, Gui et al. (2023) also found a stronger and longer-lasting decrease in soil respiration after high-severity fire compared to low-severity fire in boreal forests.

After the low severity fire, R$_{ff}$ remained similar to that at the unburnt site, and only declined significantly three to four years after the fire. We had expected to see significant reductions in R$_{ff}$ at both HM and LM in the first years after the fire. Fire can cause a significant decrease in soil microbial biomass and, thus, microbial respiration (Dooley and Treseder, 2012). After low- to moderate-severity fires across Sweden, microbial biomass decreased on average by 24% in the first year post-fire compared to unburnt stands (Eckdahl et al., 2023). Indeed, laboratory measurements of soil samples taken from our sites in 2020 confirmed these findings, showing significantly lower heterotrophic respiration at both HM and LM compared to UM (Soares et al., in review). In addition, all the burnt sites had lower concentrations of labile C (i.e. water-soluble C) than the unburnt site in the four years since the fire, suggesting reduced availability of substrates for microbial activity. Furthermore, burnt soil organic matter is more resistant to decomposition, which should also have reduced R$_{ff}$ at both HM and LM compared to UM (Pellegrini et al., 2021).

We hence conclude that the different behaviour of R$_{ff}$ after high and low severity fire at our sites likely results of changes in autotrophic respiration. The main difference between these two sites is the continuation of tree root respiration at LM (all trees survived the fire), whereas no tree root respiration occurred at HM (all trees died). In addition, the high R$_{ff}$ at LM could be due to increased tree root growth to repair roots damaged by the fire. *Pinus sylvestris* can experience significant root loss even after low-severity fire due to its shallow root distribution (Smirnova et al., 2008). Dendrometer measurements from the LM and UM sites suggest that the LM trees were allocating more C to roots after the fire compared to the UM site (Dukat et al., 2024). This may explain why LM had similar R$_{ff}$ rates to UM in the first couple of years after fire, despite declines in heterotrophic respiration. Although surface fires can cause delayed tree mortality, and thus decrease autotrophic respiration over time





(Ribeiro-Kumara et al., 2022), we did not observe any tree mortality at LM during the four years following the
fire.

In addition, there was significantly higher understory vegetation cover at LM compared to HM since the fire
(Figure 6), which would have further contributed to increasing $R_{ff}$ at LM compared to HM. Similarly, Singh et al.
(2008) found that post-fire $R_{ff}$ in boreal forests is strongly correlated with root biomass, emphasizing the
importance of vegetation regrowth and autotrophic respiration in driving post-fire $R_{ff}$.

**4.2 Effects of salvage-logging after low-severity fire on forest floor $CO_2$ fluxes**

The $R_{ff}$ was significantly lower at a site where trees that survived a low-severity fire were salvage-logged (SLM)
compared to a site where the living trees were left standing (LM; Figures 2 and 3). The removal of the living trees
stopped tree root respiration, a key component of soil respiration as discussed above, and therefore led to reduced
$R_{ff}$ at SLM compared to LM. Our results contrast with those of Kulmala et al. (2014) who observed increases in
$R_{ff}$ after a boreal forest clear-cut without fire, which they attributed to the higher soil temperatures and soil moisture
caused by the clearcut. Despite 8°C higher maximum soil temperature and higher soil moisture availability at SLM
compared to LM, this did not lead to higher $R_{ff}$ at SLM in our study. Our results thus highlight the damaging effect
of the fire and salvage-logging on $R_{ff}$ which was not temperature-limited but was instead limited by reduced
autotrophic respiration, microbial biomass and substrate availability.

The scarification of the soil at SLM also reduced $R_{ff}$. When separating our $R_{ff}$ measurements between collars
placed on areas with a remnant forest floor layer and areas where the mineral soil was exposed (Figure S2), we
found that areas with mineral soil had on average 12% lower $R_{ff}$. The areas with exposed mineral soil had low C
availability (2% C content in the mineral layer compared to 36% C content in the forest floor layer at SLM; Figures
4 and S3), which would have significantly impeded microbial activity. Similarly, in studies of the effects of soil
preparation on boreal forest soil respiration, Pumpanen et al. (2004) and Strömgren and Mjöfors (2012) measured
the lowest soil respiration in plots where bare mineral soil was exposed, which they attributed to the low organic
matter content.

It is important to note that although the salvage-logging of the living trees and soil scarification at SLM reduced
$R_{ff}$ after the fire compared to leaving the trees standing, SLM remained a net carbon source at the ecosystem level.
An eddy covariance flux tower installed at SLM showed that the site emitted an average 173 g C m$^{-2}$ per growing



season during the first four growing seasons since the fire (Kelly et al., 2024). In comparison, the living trees that
were left standing at LM were able to continue sequestering carbon at a rate of between 63-228 g C m-2 yr$^{-1}$,
despite reduced stem growth after the fire (Dukat et al., 2024).

**4.3 Effects of salvage-logging after high-severity fire on forest floor $CO_2$ fluxes**

There were no significant differences in $R_{ff}$ between the logged (SHM) and unlogged (HM) high-severity fire sites
(Figures 2 and 3). Salvage-logging of dead trees therefore appears not to have any additional impact on $R_{ff}$
compared to leaving the dead trees standing, and this did not change over the first four years since the fire.
Although salvage-logging did lead to warmer soil, this did not affect the $R_{ff}$. This could be due to the lack of
substrates available for heterotrophic respiration at both sites (as discussed in Section 4.2). The similar $R_{ff}$ at both
sites could also reflect the balance between SHM having warmer soils (which would increase $R_{ff}$) but significantly
lower understory vegetation regrowth (which would limit $R_{ff}$) whereas HM had cooler soils but higher vegetation
regrowth.

$R_{ff}$ at both sites declined during the first 3 years after the fire. We assume that this was due to a decline in
heterotrophic respiration, since autotrophic respiration could only have increased after the fire as vegetation
recolonized both sites. The reduction in heterotrophic respiration over time could result from decreased substrate
availability for microbial decomposition as any labile C and easily decomposable fine roots from the dead trees
would have been decomposed rapidly after the fire (Berg and Mcclaugherty, 2020). In addition, fire transforms
soil organic matter in multiple ways that make it harder to degrade after fire (Pellegrini et al., 2021).

Although both sites had similar $R_{ff}$, at HM the dead trees are an additional source of $CO_2$ emissions. Measurements
of respiration on dead aspen trees in a temperate forest six years after death ranged between 1 and 11 µmol $CO_2$
m$^{-2}$ s$^{-1}$ (Schmid et al., 2016), while modelled coarse woody debris respiration in a fire-affected black spruce boreal
site was on average 3 µmol m$^{-2}$ s$^{-1}$ (Bond-Lamberty et al., 2002), which is high compared with our average $R_{ff}$ of
1 µmol m$^{-2}$ s$^{-1}$ at HM. The SHM site also had planted pine seedlings that were not part of our $R_{ff}$ measurements
but contributed to the ecosystem-level carbon fluxes. Our flux tower measurements from a very similar site to
SHM (affected by the same high-severity fire in 2018 and with replanted *Pinus sylvestris* seedlings) highlighted
the importance of the planted pine seedlings in driving increases in C uptake at the site during the first four years
since the fire (HY site in Kelly et al., 2024).



### 4.4 Effects of fire and salvage-logging on forest floor CH$_4$ fluxes

The soils at all our sites were CH$_4$ sinks (Figure 3c), consuming CH$_4$ at a similar rate as reported for other Eurasian boreal forest fire sites (-1.1 to -1.3 nmol CH$_4$ m$^{-2}$ s$^{-1}$ in the first 5 years after fire; Köster et al., 2015, 2018). We did not find any effects of burn severity or salvage-logging after high-severity fire on the soil CH$_4$ fluxes in the first four years after the fire (Figure 2d, c). Our results confirm the previous findings by Kelly et al. (2021) who reasoned that the fire did not affect the mineral soil where most CH$_4$ consumption occurs, and hence did not impact the CH$_4$ fluxes. Similarly, Ribeiro-Kumara et al. (2020) found that fire had negligible effects on boreal forest soil CH$_4$ fluxes.

On the other hand, there was significantly higher CH$_4$ uptake at the salvage-logged low-severity fire site (SLM) than at the unlogged low-severity fire site (LM; Figure 2e). Although SLM had the highest SWC of all our sites (Figure 5), it also had the highest CH$_4$ uptake, which contrasts with previous findings that increasing SWC reduces CH$_4$ uptake (Smith et al., 2000). Köster et al. (2024) found that CH$_4$ uptake increased with increasing soil temperature in boreal forest soils which could explain why CH$_4$ uptake was higher at SLM since it experienced much higher soil temperatures compared to LM as a result of the salvage-logging.

### 4.5 Post-fire management effects on vegetation regrowth

At the LM and HM sites, the retention of the dead or living charred trees after the fire provided an effective source of pine seeds for natural regeneration. As a result, the unlogged sites had higher densities of pine seedlings than the salvage-logged sites, even though seeds were spread or seedlings were planted after the salvage-logging. In addition, the higher density of pine seedlings at the unlogged sites may be due to the fact that these sites had higher rates of ectomycorrhizal fungi growth compared to the salvage-logged sites (Soares et al., in review). Ectomycorrhizal fungi form symbiotic relationships with *Pinus sylvestris* trees, providing the trees with nutrients and thus ensuring healthy tree growth (Smith and Read, 2008), which may have aided the survival of the pine seedlings at the unlogged sites. We measured more than double the density of pine seedlings at LM and HM than in the low- to moderate- severity Swedish forest fire sites surveyed by Eckdahl et al. (2024). This is likely because our survey was conducted 4 years post-fire compared to 2 years in Eckdahl et al. (2024), allowing more time for seedlings to germinate. Nevertheless, the pine seedlings density at the salvage-logged sites was within the range required by Swedish law when replanting after a clear-cut (minimum 1000-1500 seedlings per ha, depending on the potential productivity of the stand; Skogsstyrelsen, 2023).





Retaining the trees at LM and HM also improved the microclimate of the forest floor by reducing soil temperature
extremes and, at HM, helping the soil retain moisture compared to the salvage-logged site (SHM; Figure 5). After
high-severity fire, the more sheltered microclimate created by retaining the dead standing trees is likely to have
contributed to the significantly higher understory vegetation cover compared to the salvage-logged high-severity
site. Several other studies have found similar results in alpine *Pinus sylvestris* stands and Mediterranean sites,
showing that salvage-logging trees after fire creates a harsher microclimate, reduces new tree seedling density and
slows the regrowth of understory vegetation (Marcolin et al., 2019; Marañón-Jiménez et al., 2011; Serrano-Ortiz
et al., 2011). However, after low-severity fire, we found that salvage-logging followed by soil scarification did not
have a significant impact on total vegetation regrowth, due to salvage-logging having opposite effects on vascular
plants (strongly negative effect) and bryophytes (positive effect; Figure 6). The fast growth of moss at the SLM
site matches previous findings that soil scarification enables the successful establishment of *Polytrichum spp.* moss
by creating areas of exposed mineral soil and reducing competition from other vegetation (Bergstedt et al., 2008).
**5 Conclusions**
We followed the recovery of boreal *Pinus sylvestris* sites during the first four years after a major forest fire in
central Sweden in 2018. A time series of measurements during these critical initial years after fire offered a unique
insight into the effects of fire severity and post-fire salvage-logging on the soil C fluxes, soil chemistry, site
microclimate as well as vegetation regrowth. The forest floor (include soil and understory vegetation) at all the
sites was a methane sink and the fire had no impact on the size of this sink. Autotrophic respiration, in particular
the presence or absence of living trees, was the main driver of differences in post-fire respiration between the sites.
Surprisingly, soil respiration continued at a similar rate for two years after low-severity fire compared to an unburnt
site. In contrast, high-severity fire or salvage-logging of living trees led to significant reductions in forest floor
respiration compared to both the unburnt and low-severity unlogged fire sites that persisted during the first four
years since the fire. Salvage-logging after high severity fire (where trees died from the fire) had no additional
impact on forest floor respiration compared to leaving the dead trees standing. However, salvage-logging did slow
the recovery of vascular vegetation and reduced the density of new *Pinus sylvestris* seedlings compared to the
unlogged sites.

Forest floor respiration at the burnt sites did not show any signs of recovery during the first four years post-fire,
and it is likely to take many more years before it reaches the levels observed at an unburnt control site due to its



tight coupling to tree root activity. Although the reduction of forest floor $CO_2$ emissions by fire and/or salvage-
logging may appear to be a positive outcome for climate change, it is important to note that our measurements
represent only part of the total ecosystem carbon balance. Our results highlight the significant and persistent
changes that occur in the soil and understory vegetation due to fire and choice of post-fire management strategy.
**Data availability**
Data will be made available via Zenodo
**Competing interests**
The authors declare that they have no conflict of interest.
**Acknowledgements**
This research was funded by the Swedish Research Council FORMAS grant 2018-02700 and the Swedish
Research Council FORMAS grant 2019-00836, the Crafoord foundation grant 20190763, Skogssällskapet Stina
Werner Fond grant 2021-094, the Royal Physiographic Society of Lund and the Swedish government through the
Strategic Research Area BECC (Biodiversity and Ecosystem Services in a Changing Climate). We are grateful to
Ellinor Delin, Niklas Båmstedt, Malin Blomberg, Jordan Mertes, Jonas Nilsson and Irene Lehner for their help
with the fieldwork. We also thank Marco Hassoldt for his ongoing support at the sites. We are grateful to Rolf
Sundell and the other forest owners for allowing us to conduct research on their land and to Jukka Kuivaniemi for
helping us establish the sites.



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
