# Peer review of "Impacts of fire severity and salvage-logging on soil carbon fluxes in a boreal forest"

_EGUsphere, 2024_

## Referee Comment (RC1)

In general, I liked the work. It is well written, it is clear, it is easy to follow and from the beginning, with Table 1 and Figure 1, the differences between the treatments are very clear, which is essential to be able to follow the results obtained later. Moreover, I agree with the authors that quantifying carbon fluxes during the initial years after fire is therefore crucial for estimating the net impact of wildfires on the carbon budget.

My main concern is that most of the conclusions of the paper were already given in a previous paper by the authors (Kelly et al. 2021) in the same sites but only one year after the fire, not four as in this case. Moreover, there is also another paper (Kelly et al. 2024) also carried out in the same fire in the same fire severities and forest management strategies 1–4 years post-fire, but in this case using eddy covariance flux towers. Therefore, my main question, given that the variations over the four years are generally not relevant to the conclusions obtained, is what does this work contribute compared to the other two mentioned above?

I also have some minor comments and doubts throughout the work:

- The recovery of the vegetation and the regeneration of the pine is added at the end without having anything to do directly with the title (which talks about soil carbon fluxes) or with the rest of the results. If it does not relate better (and not only indirectly in the end because vegetation breathes) it should not be maintained.
- In the introduction you say that the Scots pine is adapted to resist fire. I'm not sure how, it doesn't sprout, it doesn't have serotine cones, the crowns don't allow the fire to pass through without burning them. I would not make this statement.
- The design of the study is very good, with plots of different severity, and whether or not the trees are maintained. But then subsequent treatments can make the interpretation of the results difficult, both seedling planting and especially soil scarification in some cases. This issue should be discussed as a limitation of the study.
- In general, there is not a very clear temporal pattern through the four years of study, but in some cases significant increases are seen (such as that of Rff in UM in 2022 or that of SLM from 2020 to 2021) that are not fully explained and that confuse the results. The value of SWC for SLM compared to the other plots is also not evident.
- It is not clear to me that there is a need for a conclusions section like the one you currently have. Possibly some comments on the limitations found and the implications of these results on wildfire consequences on carbon fluxes would be more interesting.

---

## Referee Comment (RC2)

**Overview**

This study examines the impact of fire severity and logging/silviculture on soil greenhouse gas fluxes in the Scots pine-dominated boreal forests, which fits within the scope of BG. It utilizes the static chamber method to measure gas fluxes from forest soils over a four-year period following a large natural wildfire in Sweden (2018). The data it presents help to fill the knowledge gap about greenhouse gas fluxes in the first years after wildfire in the European boreal forests of Sweden; the emphasis is placed on $CO_2$, while $CH_4$ results are not discussed in such detail. The main findings are explained well by existing work and data, but in general, the analyses in this paper do not deepen our overall understanding of these biogeochemical processes. However, this paper has a generally good structure and appropriate references.

I see that some opportunities were missed in the data analyses that could provide insight into the drivers behind the fluxes and give more depth to the findings and discussion included here. Some more minor comments pertaining to writing style are also included below. I also have some concerns about possible sources of error.

**Main concerns**

No unmanaged control areas were included adjacent to salvage-logged and regenerated sites. At a scale of kilometers (3 km as stated in the manuscript), it allows for variation in stand and soil properties (including chemistry and biology) due to a combination of natural influences and stand management history. In this case, where burned unmanaged forests were not adjacent to salvage-logged ones and stand ages differ, some site-level description (some landscape-level details are provided, which suggests there is variation in ecology) of pre-fire ecology and management history (thinning, fertilization) to show that the sites were comparable prior to treatments would support the work. I appreciate that unburned controls are not available adjacent to burned, managed sites.

By using different gas analyzers in different years with no overlap, you are unable to separate differences between years from differences due to the analyzer (Los Gatos vs LICOR). This is a concerning source of error that has not been addressed. Any difference in the flux measured by each device should be accounted for and discussed as a source of error.

The use of mixed models to analyze fluxes is clearly described. However, when building mixed-effects models, why were the only environmental data included soil temperature (for $R_{ff}$) and moisture (for $CH_4$)? Expanding the models to include soil chemistry measurements and vegetation cover would allow you to use your own measurements to address sources of variation in the measured fluxes, which is included in the discussion. At least in L441-442 and L497-502 it is mentioned that there is a causal effect between vegetation, soil chemistry and fluxes, so why not test it with your own data? This would improve the interpretation of the results and the depth of the discussion. If these were tested, but no significant impact was found, it also would provide value for the reader to know this.

While the use of beta regression is appropriate for the proportional cover data, why were multivariate techniques not used to assess changes in vegetation over time? This could strengthen the understanding of the nature of the changes and differences in plant communities between sites. For example, NMDS and adonis could be applied to test differences in communities between sites and years. Why was vegetation measured at species-level if only overall cover was considered?

On the SLM site, half the collars were on furrows, while the other half were on ridges. By pooling these to represent the whole site, it assumes that each of these surface types covers exactly half of the sites. Is this accurate, or would weighting the fluxes by actual precent cover of each surface type for pooling the data provide a more accurate analyses of the site? Please at least report the actual percent cover of each surface type (furrow, ridge) on the site. If this is much different than half-and half, the fluxes of each surface type should be weighted by proportional cover for pooling. This comment is not relevant if there is no significant difference between the fluxes on ridges and furrows – was this tested? If so, it should be reported. Addendum: I have now found this in the supplement (please reference this so the reader knows where to look), and it looks like $CH_4$ differs significantly between scarified and unscarified surfaces, which raises the question of whether these differences are coming from the removal of the vegetation or from disturbing the soil.

Figure captions should provide enough information for figures to be easily interpreted without reference to the text. Please go through the captions (including supplement) and ensure that this is the case for all the figures. number of replicates? years? sensors?

Figure 5. What year are these data from? Please update the caption to include this and which sensors the data are from. Why are the data from only one year presented if you have data from the whole study period? Data from multiple years would help to understand the variation between the years, which may have impacted fluxes.

Section 4.5. Your aims and hypotheses focus on fluxes, so why are you reporting and discussing this in such detail? I know that vegetation can help to explain fluxes, but here, vegetation is not included as a predictor of fluxes in your models so it becomes ancillary, not directly adding to the main story. Somehow, this section doesn't quite fit with the way the research is structured and the data are currently analyzed.

The conclusions section is redundant.

**Minor comments**

Beware capitalization rules (e.g. L96 podzol, gas analyzer)

The word "data" is almost always plural, i.e. "data **are**," unless you are talking about a single point.

Remove unnecessary gridlines in tables.

Be aware of correct hyphenation. "Mixed-effects models" or "mixed models." When used as an adjective, "low-severity" and "high-severity" is correct, be consistent.

When introducing the sites, I don't like the usage of the wording "groups," which makes it seem like these are actual groups of unique sites. Consider a different term. These "groups" seem to be more akin to comparison sets than actual groups of sites (L106-115; L167-169). Comparisons were only done within groups, not between them, making them comparison sets, i.e. group 2 contains a site from group 1, but groups 1 and 2 are not compared.

The methods section lacks a parallel structure. Most data analyses are grouped in the same section as the methods for collecting those data. However, flux analyses are given their own section. The separate section for flux analyses can be eliminated (other data analyses are described under the same section as the relevant data collection) or consider putting all data analyses in a separate section.

Tables 2,3, and 4 could be moved to supplementary information. The results of these ANOVA/Chi-squared analyses (significance) are already indicated by the letters on the boxes in Figs. 2 and 6.

**Lines**

L48: Currently reads that soil respiration leads to high tree mortality? Please fix this sentence.

L87-88: you don't actually use these data to explain fluxes - they are analyzed separately.

L98: there should be a clearer definition of severity as it relates to soils, especially for high-severity areas (was there any complete consumption of the upper organic layers?). tree mortality ok.

Table 1 (L128): is very informative for detailing the experimental design, but format to remove unnecessary grid lines. Please show *n* (in the caption at least) for forest floor organic/charred depth measurements.

Fig. 1 (L134): align UM pic with other 2019 photos

L147-148: beware of capitalization of common nouns. What is the model of the Los Gatos analyzer? GLA132-GGA?

L156: min = minute, use consistent unit abbreviations (see "second" on L157)

L 203: What were the 4 composite samples per site? Was each composite sample a different layer (i.e. four layers, and all the material from that layer in each sample was pooled)? Or did the layers get pooled together? It's not clear to me. Or were they pooled so that each transect had 2 pooled samples at the end: one organic/forest floor, the other mineral?

L208: reference? data?

L225: The way the methods are currently worded, it is unclear if the vegetation surveys took place annually or took 3 years to complete. This can be inferred as annual surveys from Fig. 6 but is not immediately evident.

L247-248: "lower but not as low as" is awkward. Suggestion: "intermediate."

L 263-265: Reporting each value is tedious. These values are already presented in Fig. 2 (d-f). A supplementary table with exact values could be included if it is necessary to include precisely these.

L268-269: Does this also refer to Fig 2?

L273: Fig.2 "...where letters above the **boxes**..."

L275: This period should be a comma? add the word "and"

Fig 3 (L281): The legend needs to be simplified. Sites are listed multiple times.

Fig. 4. (L321-344): Please check the calculations for these points in regards to the error. Why are the SE error bars so small and nearly non-existent for some data points? Also, reference to the wrong supplementary figure.

Fig. 6. (L417): Why is the unburned control not included here? You need to include this, otherwise there is no "baseline" control. Panels d-f show that the change in cover over time for some sites appears to be significant. In this case, is it meaningful to group several years together in panels a-c (panels d-f already show which sites have higher cover in addition to annual changes)? Even better, these analyses can be replaced with multivariate methods.

L428, L458 etc., no need to refer to results figures in the discussion

467-468: Was this relationship tested and found not to be causal (i.e. that you didn't find any controls of temperature/moisture on Rff?), or was this an observation that temp/moist was higher, but not Rff?

L504-512: I'm not sure that this is necessary – how does it serve to discuss/explain the result that was found?

Supplement

I don't understand how Fig. S1, Table S1, and Table S2 improve our understanding of the data compared to what is already presented in the results. I see it covers a different time period (July/Aug vs entire growing season).

Fig. S1 Also, the legend needs to be simplified; sites are listed multiple times.

Fig S4 - year? sensor?

---

## Author Comment (AC1)

**Response to reviewer 2:**

Thank you for taking the time to read our manuscript so thoroughly. Your comments have greatly improved the quality of the text. We have provided a point-by-point response (in blue) to your comments (in black):

**Overview**

This study examines the impact of fire severity and logging/silviculture on soil greenhouse gas fluxes in the Scots pine-dominated boreal forests, which fits within the scope of BG. It utilizes the static chamber method to measure gas fluxes from forest soils over a four-year period following a large natural wildfire in Sweden (2018). The data it presents help to fill the knowledge gap about greenhouse gas fluxes in the first years after wildfire in the European boreal forests of Sweden; the emphasis is placed on $CO_2$, while $CH_4$ results are not discussed in such detail. The main findings are explained well by existing work and data, but in general, the analyses in this paper do not deepen our overall understanding of these biogeochemical processes. However, this paper has a generally good structure and appropriate references.

I see that some opportunities were missed in the data analyses that could provide insight into the drivers behind the fluxes and give more depth to the findings and discussion included here. Some more minor comments pertaining to writing style are also included below. I also have some concerns about possible sources of error.

**Main concerns**

No unmanaged control areas were included adjacent to salvage-logged and regenerated sites. At a scale of kilometers (3 km as stated in the manuscript), it allows for variation in stand and soil properties (including chemistry and biology) due to a combination of natural influences and stand management history. In this case, where burned unmanaged forests were not adjacent to salvage-logged ones and stand ages differ, some site-level description (some landscape-level details are provided, which suggests there is variation in ecology) of pre-fire ecology and management history (thinning, fertilization) to show that the sites were comparable prior to treatments would support the work. I appreciate that unburned controls are not available adjacent to burned, managed sites.

We fully understand Reviewer 2's concerns but would like to highlight that all the sites except SLM were adjacent to each other (within <1000m), except SLM that was 3km away. The current study is based on a 'natural' experiment which we could not control nor influence: the wildfire swept through the forests with different fire-severities and the forest plots were owned by multiple private owners who individually decided about the post-fire forest management of their plots. This setup offered a unique opportunity, but also came with some limitations. Detailed management information from the sites before the fire is, unfortunately, not available. Since we only started our measurements at the sites after the fire, we do not have pre-fire ecological information at the sites. Instead, we assume that our unburnt site represents the pre-fire ecological conditions of our burnt sites. We will add the following text to the methods section to clarify this point:

*"Since we do not have pre-fire measurements at these sites, we assume that the unburnt site (UM) is representative of the pre-fire conditions at the sites."*

We will also add a 'Limitations' section at the end of the discussion which includes information about the differences in the SLM site compared to the other sites:

*"The SLM site was located 3 km away from the other sites on loamy soil which increased the SWC at that site compared to all the others which were on sandy soils. In addition, furrows with exposed mineral soil at SLM retained more water and thus had higher SWC (data not shown) than all the other sites where the organic layer remained (which when burnt can become hydrophobic and retains less water; Certini 2005). Although these conditions impaired the comparison between SLM and LM slightly, our analysis of the $R_{ff}$ data and vegetation cover still shows a significant impact of salvage-logging of living trees at SLM"*

By using different gas analyzers in different years with no overlap, you are unable to separate differences between years from differences due to the analyzer (Los Gatos vs LICOR). This is a concerning source of error that has not been addressed. Any difference in the flux measured by each device should be accounted for and discussed as a source of error.

We agree that it is not ideal to have measurements from different analysers in different years. We used the same chamber for all measurements across all years which helped minimize the differences in fluxes measured by each analyser. Both analysers were calibrated before the fieldwork campaigns. In addition, Pumpanen et al. (2004) showed there was no significant difference in the $CO_2$ fluxes between different closed dynamic chambers and gas-analyser systems like ours. We have clarified this point and included information on the gas analyser uncertainty in the methods section 2.2 by adding the following text:

*"Using different analysers does introduce some uncertainty in the comparison of the fluxes from different years. However, the analysers were calibrated before the sampling rounds and the same chamber was used for all measurements in all years, which will have reduced this uncertainty. Furthermore, previous work has shown no significant differences in flux measurements between different models of closed dynamic chambers and gas analysers (Pumpanen et al., 2004). Both analysers recorded data at 1 Hz and offered high accuracy for both $CO_2$ and $CH_4$ concentration measurements."*

The use of mixed models to analyze fluxes is clearly described. However, when building mixed-effects models, why were the only environmental data included soil temperature (for $R_{ff}$) and moisture (for $CH_4$)? Expanding the models to include soil chemistry measurements and vegetation cover would allow you to use your own measurements to address sources of variation in the measured fluxes, which is included in the discussion. At least in L441-442 and L497-502 it is mentioned that there is a causal effect between vegetation, soil chemistry and fluxes, so why not test it with your own data? This would improve the interpretation of the results and the depth of the discussion. If these were tested, but no significant impact was found, it also would provide value for the reader to know this.

We agree that including the soil chemistry and vegetation cover data in the mixed models would support the interpretation of our findings. However, both the vegetation survey and the soil sampling for the chemistry analysis were conducted less frequently (once per year) than the soil carbon flux measurements (monthly during the growing season) and at different locations within each site. Therefore, it is not possible to directly link a specific soil sample or vegetation survey quadrat with a soil flux measurement, which would be needed to include the chemistry and vegetation data in the mixed-effects models.

Instead, in the revised manuscript we will include a new correlation analysis that assesses the relationship between the soil fluxes, chemistry and vegetation cover data. This analysis pools data from all available sites and years (n = 14). Although this leads to an issue of autocorrelation and pseudo-replication (multiple measurements from each site), it allows us to assess the relationship between all of these variables. If we would conduct an analysis separately for each site, we would only have 2-4 data points (maximum 4 years of soil chemistry data available, for UM only 2 years of vegetation survey data available). This is too small to draw any meaningful conclusions. We performed the analysis using spearman's rank since some of the variables were not normally distributed and we have not included p-values since their reliability is questionable due to the autocorrelation/pseudo-replication. Nevertheless, having the correlation matrix gives us an insight into the direction and strength of the relationships between the soil fluxes, vegetation cover and chemistry variables and help us strengthen the discussion.

The following new figure will be added to the manuscript:

[Figure]

*Figure 7. Spearman's correlation coefficients of soil carbon fluxes, vegetation cover and soil chemistry (from the forest floor layer), using annually averaged variables for all available years (2020-2022 for all sites except UM, 2021-2022) and all sites (n = 14). The size of the circles is proportional to the absolute value of the correlations.*

The findings from this analysis will also be added to the discussion in the following places:

*"In addition, all the burnt sites had lower concentrations of labile C (i.e. water-soluble C) than the unburnt site in the four years since the fire, and labile C concentration was positively related*

*to $R_{ff}$ (Figure 7). This suggests that the reduced availability of substrates for microbial activity at the burnt sites could have reduced their $R_{ff}$."*

*"In addition, we found significantly faster recovery of the understory vegetation cover at LM compared to HM since the fire (Figure 6) and total vegetation cover was strongly positively associated with $R_{ff}$ (Figure 7). This would have further contributed to increasing $R_{ff}$ at LM compared to HM. [...] Furthermore, vascular cover was very strongly and positively associated with labile C concentration, indicating that the regrowth of vascular vegetation after fire was important for increasing the availability of root substrates which in turn would have contributed to the heterotrophic component of $R_{ff}$."*

While the use of beta regression is appropriate for the proportional cover data, why were multivariate techniques not used to assess changes in vegetation over time? This could strengthen the understanding of the nature of the changes and differences in plant communities between sites. For example, NMDS and adonis could be applied to test differences in communities between sites and years. Why was vegetation measured at species-level if only overall cover was considered?

When we decided on a vegetation sampling strategy we decided to collect as much data as was feasible to allow tracking changes in individual species over time as this may be useful for future work. However, it turned out that only a very few species were represented in the vascular plant communities, and total cover per plot remained low even in the third year. This was even clearer regarding bryophytes. Therefore, we decided not to attempt to analyse the community composition in more detail, but instead focused on analysing the cover of vascular plants and bryophytes separately, as these have different ecological roles in boreal forests.

On the SLM site, half the collars were on furrows, while the other half were on ridges. By pooling these to represent the whole site, it assumes that each of these surface types covers exactly half of the sites. Is this accurate, or would weighting the fluxes by actual precent cover of each surface type for pooling the data provide a more accurate analyses of the site? Please at least report the actual percent cover of each surface type (furrow, ridge) on the site. If this is much different than half-and half, the fluxes of each surface type should be weighted by proportional cover for pooling. This comment is not relevant if there is no significant difference between the fluxes on ridges and furrows – was this tested? If so, it should be reported. Addendum: I have now found this in the supplement (please reference this so the reader knows where to look), and it looks like $CH_4$ differs significantly between scarified and unscarified surfaces, which raises the question of whether these differences are coming from the removal of the vegetation or from disturbing the soil.

Good point. Based on a visual analysis of drone images from SLM, we estimated that 66% of the site is covered by ridges while 33% is covered by furrows. Our placement of the collars (7 on ridges and 5 in furrows) therefore accurately reflects the proportion of the site covered by each land type. We will add the following text in the Methods Section 2.2 to clarify this:

*"This distribution of the collars reflects the areal proportion of the ridges and furrows at SLM that cover about two-thirds and one-third of the site, respectively"*

Figure S2, which shows the fluxes separately for the ridges and furrows, is already referenced in the Methods Section 2.2. We will add references to it in the results Section 3.2 and in the discussion Section 4.4. We will also update Figure S2 and have tested for significant differences in the $R_{ff}$ and $CH_4$ fluxes between ridge and furrow collars using a mixed model approach similar to the main flux analysis. The new Figure S2 is shown below:

[Figure]

*Figure S2. (a) Growing season mean (±SE) forest floor respiration ($R_{ff}$) and (b) $CH_4$ fluxes and (c) boxplots of all $R_{ff}$ and (d) $CH_4$ flux measurements, at SLM (low-severity fire, living trees salvage-logged). The data was divided into collars on ridges where soil organic layer is intact ('organic', n = 7) and collars in furrows where the mineral soil is exposed due to the scarification ('mineral', n = 5). Different letters above the boxplots indicate significant differences and triangles indicate the mean flux. Boxplot whiskers show the minimum and maximum flux.*

We will add the following text to the discussion section 4.4 to clarify the effects of soil scarification on the $CH_4$ flux at SLM:

*"The soil scarification at SLM also led to significant differences in $CH_4$ uptake within the site, with higher uptake in the ridges of remaining organic soil than in the furrows with exposed mineral soil (Figure S2). Our observations agree with findings from $CH_4$ measurements in agricultural soils which show that no-till soils can have significantly higher $CH_4$ uptake than tilled soils (Prajapati and Jacinthe, 2014). Tilling destroys the soil structure and reduces soil gas diffusivity, thus limiting $CH_4$ uptake. The same process likely also occurred during the soil scarification, reducing $CH_4$ uptake in the furrows."*

Figure captions should provide enough information for figures to be easily interpreted without reference to the text. Please go through the captions (including supplement) and ensure that this is the case for all the figures. number of replicates? years? sensors?

Thank you for the suggestion. We will go through all figure captions including in the SI and add the relevant information where necessary

Figure 5. What year are these data from? Please update the caption to include this and which sensors the data are from. Why are the data from only one year presented if you have data from the whole study period? Data from multiple years would help to understand the variation between the years, which may have impacted fluxes.

Thank you for pointing out the missing information. We will update the figure caption with the following text:

*All data are from 2022 from TOMST TMS-4 sensors.*

Since the TMS-4 sensors were only installed in spring 2022, we only have data from that year for all the sites. Air temperature, soil temperature and SWC data is available since 2019 from other sensors but only at UM, LM and SLM. We will add a figure with these data in the SI (Figure S5, see below). In the main manuscript, we focus on presenting the 2022 TMS-4 data which includes all the sites. We will add text to the methods Section 2.5 to clarify this point:

*"We only present the data from the TOMST TMS-4 loggers from 2022 in the main text because this is the only sensor and year from which we have data for all the sites. However, the longer-term data of air temperature, soil temperature and SWC from UM, LM and SLM are shown in Figure S5."*

[Figure]

*Figure S5. Daily mean (a) air temperature (Tair), (b) soil temperature at 7.5 cm depth (Tsoil) and (c) soil water content at 7.5 cm depth (SWC) for 2019-2022. Tair was measured at 2 m height using a HygroVUE10 (Campbell Scientific, Inc). Tsoil is the average from three CS655 sensors and one 107 Thermistor (all from Campbell Scientific, Inc) per site. SWC is the average from three CS655 sensors per site.*

Section 4.5. Your aims and hypotheses focus on fluxes, so why are you reporting and discussing this in such detail? I know that vegetation can help to explain fluxes, but here, vegetation is not included as a predictor of fluxes in your models so it becomes ancillary, not directly adding to the main story. Somehow, this section doesn't quite fit with the way the research is structured and the data are currently analyzed.

Thank you for raising this issue, we will remove this section from the revised manuscript.

The conclusions section is redundant.

Biogeosciences requires a conclusions section therefore we will keep this section.

**Minor comments**

Beware capitalization rules (e.g. L96 podzol, gas analyzer)

Thank you, we will ensure that capitalization rules are followed throughout the text.

The word "data" is almost always plural, i.e. "data **are,**" unless you are talking about a single point.

We will make sure that 'data' is used correctly throughout the manuscript.

Remove unnecessary gridlines in tables.

We will make sure to follow the formatting suggestions of Biogeosciences for all tables.

Be aware of correct hyphenation. "Mixed-effects models" or "mixed models." When used as an adjective, "low-severity" and "high-severity" is correct, be consistent.

We will ensure that correct hyphenation is used throughout the manuscript.

When introducing the sites, I don't like the usage of the wording "groups," which makes it seem like these are actual groups of unique sites. Consider a different term. These "groups" seem to be more akin to comparison sets than actual groups of sites (L106-115; L167-169). Comparisons were only done within groups, i.e. "group 2" contains a site from "group 1", but "groups" 1 and 2 are not compared.

We agree that this is not an ideal term to describe how the sites are grouped together. However, we will stick to using the term 'group' since it is easy to understand and use but we will add text to the methods to explain that the same site can occur in multiple groups.

The methods section lacks a parallel structure. Most data analyses are grouped in the same section as the methods for collecting those data. However, flux analyses are given their own section. The separate section for flux analyses can be eliminated (other data analyses are described under the same section as the relevant data collection) or consider putting all data analyses in a separate section.

Thank you for pointing out this issue. We will change the methods section so that the soil flux data analysis is included as a subsection within the soil flux section.

Tables 2,3, and 4 could be moved to supplementary information. The results of these ANOVA/Chi-squared analyses (significance) are already indicated by the letters on the boxes in Figs. 2 and 6.

Thank you for this suggestion. We prefer to keep these tables in the main text because they provide an easy-to-read overview of all the model results and model fit. For the burn severity

site group, the letters in Figures 2 and 6 are from Tukey post-hoc tests which are not included the Tables.

**Lines**

L48: Currently reads that soil respiration leads to high tree mortality? Please fix this sentence.

Will do.

L87-88: you don't actually use these data to explain fluxes - they are analyzed separately.

We will add a correlation analysis, as described in our answer to a previous comment, that helps identify which of the soil chemistry and vegetation survey data are associated with changes in the soil fluxes. All the data are also referred to in the discussion in order to explain the patterns observed in the soil fluxes.

L98: there should be a clearer definition of severity as it relates to soils, especially for high-severity areas (was there any complete consumption of the upper organic layers?). tree mortality ok.

We will add the following clarification:

*"The burnt area included areas affected by high-severity and low-severity fire. Our definition of fire severity is based on the impact of the fire on the trees. We define high-severity fire as crowning fire leading to complete tree mortality, almost complete combustion of the understory vegetation and substantial consumption of the soil organic layer, whereas low-severity fire was defined as surface fire that almost all trees survived but which burnt most of the soil organic layer and understory vegetation. Table 1 includes data on the thickness of the forest floor at each site"*

Table 1 (L128): is very informative for detailing the experimental design, but format to remove unnecessary grid lines. Please show *n* (in the caption at least) for forest floor organic/charred depth measurements.

Will do.

Fig. 1 (L134): align UM pic with other 2019 photos

We will add a photo from UM in 2022 and add the new Figure (below) to the revised manuscript:

[Figure]

L147-148: beware of capitalization of common nouns. What is the model of the Los Gatos analyzer? GLA132-GGA?

We will ensure all the text follows capitalization rules. We will add the model information of the gas analyser to the text (model 915-0011).

L156: min = minute, use consistent unit abbreviations (see "second" on L157)

Will do.

L203: What were the 4 composite samples per site? Was each composite sample a different layer (i.e. four layers, and all the material from that layer in each sample was pooled)? Or did the layers get pooled together? It's not clear to me. Or were they pooled so that each transect had 2 pooled samples at the end: one organic/forest floor, the other mineral?

We will clarify this by adding the following text:

*"The 20 samples of the forest floor layer were pooled to create four composite samples per site, and the same process was repeated for the mineral soil samples (see Kelly et al., 2021 for details)."*

L208: reference? data?

We will clarify the sentence as follows:

*"No carbonates are present in the lithology of the study area, and no substantial amount of carbonates were found in the charred forest floor in the months following the fire (data not shown), so all soil carbon is assumed to be organic"*

Due to the acidic forest floor (pH between 3.5 and 4.5) at our sites, we can assume that if any carbonates were formed by combustion during the fire, that they disappeared shortly after the fire.

L225: The way the methods are currently worded, it is unclear if the vegetation surveys took place annually or took 3 years to complete. This can be inferred as annual surveys from Fig. 6 but is not immediately evident.

Yes, correct, the surveys were done on an annual basis. We will clarify this in the text as follows:

*"We surveyed the coverage of the understory vegetation at the burnt sites yearly in July 2020-2022 (unburnt site only in 2021-2022)."*

L247-248: "lower but not as low as" is awkward. Suggestion: "intermediate."

We will clarify this text.

L 263-265: Reporting each value is tedious. These values are already presented in Fig. 2 (d-f). A supplementary table with exact values could be included if it is necessary to include precisely these.

We agree that this is not the most elegant sentence. However, we think that some readers of the manuscript will be happy that we reported the exact values because it will make it easier for them to compare our measurements with theirs without having to go digging in the SI.

L268-269: Does this also refer to Fig 2?

No, it refers to Table 3. We will add a reference to Table 3 at the end of the sentence.

L273: Fig.2 "…where letters above the **boxes**…"

Will do.

L275: This period should be a comma? add the word "and"

Will do.

Fig 3 (L281): The legend needs to be simplified. Sites are listed multiple times.

We will simplify the legend as follows:

[Figure]

Fig. 4. (L321-344): Please check the calculations for these points in regards to the error. Why are the SE error bars so small and nearly non-existent for some data points? Also, reference to the wrong supplementary figure.

Thank you for spotting this issue, there was an error in the SE calculation for some of the subplots and we will fix this in the revised manuscript (see new figure below). The change does not affect the results or conclusions. We will also correct the reference to point to Figure S3.

[Figure]

Fig. 6. (L417): Why is the unburned control not included here? You need to include this, otherwise there is no "baseline" control. Panels d-f show that the change in cover over time for some sites appears to be significant. In this case, is it meaningful to group several years together in panels a-c (panels d-f already show which sites have higher cover in addition to annual changes)? Even better, these analyses can be replaced with multivariate methods.

Thank you for this suggestion. We did not include the UM control site because it was not included in the statistical analysis (fewer years of data available compared to the burnt sites). However, we will change the figure to include the UM site. We will keep subpanels a-c because it gives an easy-to-read overview of the results of the statistical differences in vegetation cover between the sites. The beta regressions tested for differences in the mean cover using all years of data combined therefore it is meaningful to plot this data.

[Figure]

*Figure 6. Mean ± SE of raw data from 2020-2022 combined (UM only 2021-2022) of (a) total vegetation cover, (b) vascular cover and (c) bryophyte cover in the understory. SHM mean and SE of bryophyte cover were <1% and are not visible on the plot. P-values show results of chi-square tests for significant differences between sites within each group (LM vs HM, LM vs SLM and HM vs SHM; Table 4). (d-f) annual mean ± SE of raw total vegetation, vascular or bryophyte cover, respectively.  Site characteristics: LM (low-severity fire), HM (high-severity fire, dead trees left standing), living trees left standing), SLM (low-severity fire, living trees salvage-logged) and SHM (high-severity fire, dead trees salvage-logged).*

We will also add text to the results Section 3.4 that discusses the variability of the vegetation over time (see below). Note that in the discussion, we do not include any reference to the changes in vegetation over time, but only to the differences between the sites which are consistent over time.

*"UM was not included in the statistical analysis of the fire severity group because there were fewer years of data available at UM; for reference it is plotted in Figure 6. Total vegetation cover, and in particular bryophyte cover, increased at UM between 2021 and 2022 (Figure 6d, e). The large variability vegetation cover between the two years at UM suggests that the variation over time in vegetation at the other sites may be partly due to factors other than the fire or post-fire management. Such factors could include the patchiness of the vegetation and differences in moisture availability (e.g. wet conditions proceeding the vegetation survey in 2022 may have made the bryophytes appear larger since they were full of water whereas in 2021 conditions were much drier). The latter may explain why vegetation cover at SLM followed similar temporal trends as at UM, since SLM had the highest bryophyte cover of any of the burnt sites (Figures 6d-f). However, the differences in mean total, vascular or bryophyte cover between the sites in each group were persistent across the years."*

L428, L458 etc., no need to refer to results figures in the discussion

We think that this is a matter of personal taste and prefer to keep these references.

467-468: Was this relationship tested and found not to be causal (i.e. that you didn't find any controls of temperature/moisture on Rff?), or was this an observation that temp/moist was higher, but not Rff?

This was an observation but we also tested the temperature control on $R_{ff}$ in the mixed models and found it to be significant. We will clarify this point by adding the following text to the revised manuscript:

*"Despite 8°C higher maximum soil temperature and higher soil moisture availability at SLM compared to LM, and the significant effect of soil temperature on $R_{ff}$ (Table 2), this did not lead to higher $R_{ff}$ at SLM than LM in our study."*

L504-512: I'm not sure that this is necessary – how does it serve to discuss/explain the result that was found?

It helps put the $R_{ff}$ results into context. It is important to explain that our $R_{ff}$ measurements do not include certain sources or sinks of carbon that could have an important impact on the ecosystem carbon budget.

Supplement

I don't understand how Fig. S1, Table S1, and Table S2 improve our understanding of the data compared to what is already presented in the results. I see it covers a different time period (July/Aug vs entire growing season).

These figures and tables help to show that the difference in sampling period between years does not affect the interpretation of our results. In 2022 measurements were only taken in July and August but in 2019-2021, measurements were done monthly between June-September. This is already discussed in Section 2.2 in the main text:

*"The difference in the length of the sampling period had little effect on the soil greenhouse gas results: an analysis using only July-August data from all years (Figure S1 and Tables S1 and S2) showed the same trends in the $R_{ff}$ data and only minor differences in the $CH_4$ data as when the June-September data was included (Figure 3a, 3c and Tables 2 and 3)."*

Fig. S1 Also, the legend needs to be simplified; sites are listed multiple times.

We will simplify this legend as follows:

[Figure]

[Figure]

Fig S4 - year? sensor?

We will add this information to the caption.

---

## Author Comment (AC2)

**Response to reviewer 1:**

Thank you for taking the time to read our manuscript. Your comments have greatly improved the quality of the text. We have provided a point-by-point response (in blue) to your comments (in black):

 In general, I liked the work. It is well written, it is clear, it is easy to follow and from the beginning, with Table 1 and Figure 1, the differences between the treatments are very clear, which is essential to be able to follow the results obtained later. Moreover, I agree with the authors that quantifying carbon fluxes during the initial years after fire is therefore crucial for estimating the net impact of wildfires on the carbon budget.

Thank you for your positive feedback, we appreciate it.

My main concern is that most of the conclusions of the paper were already given in a previous paper by the authors (Kelly et al. 2021) in the same sites but only one year after the fire, not four as in this case. Moreover, there is also another paper (Kelly et al. 2024) also carried out in the same fire in the same fire severities and forest management strategies 1–4 years post-fire, but in this case using eddy covariance flux towers. Therefore, my main question, given that the variations over the four years are generally not relevant to the conclusions obtained, is what does this work contribute compared to the other two mentioned above?

This work contributes two major new points compared to the previous two papers at the Ljusdal fire sites reviewer 1 has referred to:

1. The introduction and analysis of soil fluxes at a new site (SLM) that allowed us, for the first time, to explicitly assess the impacts of salvage-logging after a low-severity fire compared to leaving the living trees standing on the soil carbon fluxes. This was not possible in Kelly et al. (2021), since we did not collect data from the SLM site in the first year after the fire. Although the SLM site was featured in the eddy covariance flux tower paper (Kelly et al., 2024), that analysis compared ecosystem flux data at two burnt stands. In the present study, we compare soil flux data at five burnt and unburnt stands with different post-forest management approaches. The inclusion of the SLM site is one of the most interesting and important aspects of the current manuscript. Low-severity fires are the most common type of fire in the Eurasian boreal forest and thus it is vital to understand how salvage-logging of live after these fires (a common post-fire management approach in Scandinavia) affects the ecosystem carbon balance.

2. The Kelly et al. (2021) paper only presented one year of soil flux data from our sites in the first year after the fire. It was therefore only a snapshot of how the sites were recovering after the fire. By presenting a time series of four years of data in the present manuscript, we could confirm our previous findings and highlight how long it is taking the sites to recover, since no major changes in the fluxes have occurred since the first year post-fire. The recovery time of a forest after natural and/or human-induced disturbance is a highly topical issue, and we see this study as an important contribution to the debate. We will emphasize this point in the revised manuscript by adding the following new section to the discussion:

*"**4.4 No recovery of $R_{ff}$ four years after fire**

*By the fourth year after the fire, $R_{ff}$ and total understory vegetation cover was still substantially lower at all the burnt sites compared to the unburnt site. These differences were largest after high-severity fire and/or salvage-logging. None of the site groups we tested showed positive trends in $R_{ff}$ over time since the fire, indicating it may take many more years until $R_{ff}$ recovers to pre-fire levels. Parro et al. (2019) found no significant difference in $R_{ff}$ between 5 or 21 years after fire in Estonian Pinus sylvestris forests on sandy soils similar to our sites, and suggest that two decades may not provide sufficient time for $R_{ff}$ to recover in such low fertility sites. Similarly, in their review of boreal forest $R_{ff}$ fluxes after fire, Ribeiro-Kumara et al. (2020) found that it took between 10 to 30 years for $R_{ff}$ to recover after fire. Since we have shown that tree respiration is such an important driver of $R_{ff}$, the recovery time of $R_{ff}$ will likely be tightly coupled to the time it takes for trees to regrow or recover from fire-related injuries, which in turn is linked to how the sites were managed after the fire (salvage-logged versus unlogged, planted seedlings or seeds sown)."*

The recovery of the vegetation and the regeneration of the pine is added at the end without having anything to do directly with the title (which talks about soil carbon fluxes) or with the rest of the results. If it does not relate better (and not only indirectly in the end because vegetation breathes) it should not be maintained.

Good point, we will remove this section in the revised manuscript.

In the introduction you say that the Scots pine is adapted to resist fire. I'm not sure how, it doesn't sprout, it doesn't have serotine cones, the crowns don't allow the fire to pass through without burning them. I would not make this statement.

Rogers et al. (2015) describes Scots pine as "resisting" fire because it is adapted to survive low-severity fires and prevent them from spreading into the tree canopy. Adaptations include thick bark and loss of lower branches. Trees that sprout and have serotine cones are called 'fire embracers' by Rogers et al. (2015) since they need fire to reproduce.

We will change the wording in the introduction to clarify this point as follows:

*"In boreal Eurasia, forests include tree species such as larch and Scots pine that are adapted to survive low-severity fire and prevent it from spreading into the forest canopy (Rogers et al. 2015)."*

The design of the study is very good, with plots of different severity, and whether or not the trees are maintained. But then subsequent treatments can make the interpretation of the results difficult, both seedling planting and especially soil scarification in some cases. This issue should be discussed as a limitation of the study.

Thank you for your positive feedback. We agree that the difference in the post-fire management treatments between some of the sites is not ideal. However, as already explained in the methods, we do not compare the SHM and SLM sites because of these differences in their postfire management. In the revised version of the manuscript we will add a 'Limitations' section to the discussion that clarifies this point:

*"4.6 Limitations*

*Our study is based on an opportunistic design as we could not control nor influence the wildfire or the post-fire forest treatments. The wildfire burnt the study sites at different severities and the private owners of those sites independently decided which post-fire management approach to follow. Despite the inherent limitations of such a design, it did offer a unique opportunity as the UM, LM, HM, and SHM sites were all comparable and within less within less than 1000 m of each other. However, it was not possible to analyse the interaction between fire severity and salvage-logging because the SHM and SLM sites were treated differently after they were salvage-logged (i.e. pine seedlings planted at SHM versus soil scarification and spreading of pine seeds at SLM). We also did not have an unburnt clear-cut site that would have made a full factorial design and allowed us to separate the effects of the salvage-logging and the fire. Future work at other sites should investigate these effects, their interaction and should compare burnt sites with and without soil scarification to better distinguish between the effects of the fire and post-fire treatment of the soil on the forest recovery."*

In general, there is not a very clear temporal pattern through the four years of study, but in some cases significant increases are seen (such as that of Rff in UM in 2022 or that of SLM from 2020 to 2021) that are not fully explained and that confuse the results. The value of SWC for SLM compared to the other plots is also not evident.

We agree that the high $R_{ff}$ in UM in 2022 is confusing. However, we have looked through our photos from the collars, the timing of the measurements, the raw data from the gas analyser and the soil moisture and soil temperature data and cannot find any errors and abnormalities in the measurements. We have therefore decided to keep these measurements in the analysis. Our decision is already explained in the results section 3.1.

*"$R_{ff}$ at UM was much higher in 2022 compared to previous years. The high $R_{ff}$ values at UM in 2022 were driven by a few measurements of very high $R_{ff}$ in August 2022. We could not find any fault with the measurements and therefore retained them in the analysis."*

Reviewer 1 refers to "significant increases are seen [...] SLM from 2020 to 2021", which we believe refers to the decline in the $CH_4$ uptake at SLM (there is no change in SLM $R_{ff}$ during that period). In the revised manuscript, we will add the following text to acknowledge this result in the discussion. Unfortunately, we have not been able to identify what is causing this decrease:

*"Although $CH_4$ uptake decreased significantly between 2020 and 2021 at SLM, this change was not related to changes in soil moisture or temperature conditions because they were similar in both years. In addition, there were only weak correlations between $CH_4$ flux and all the soil chemistry variables. We have not been able to identify the cause of the changes in the $CH_4$ fluxes over time at the SLM site."*

We have also added a new 'Limitations' section to the discussion which includes information on the high SWC at SLM compared to the other sites:

*"The SLM site was located 3 km away from the other sites on loamy soil which increased the SWC at that site compared to all the others which were on sandy soils. In addition, furrows with exposed mineral soil at SLM retained more water and thus had higher SWC (data not shown) than all the other sites where the organic layer remained (which when burnt can become hydrophobic and retains less water; Certini, 2005). Although these conditions impaired the comparison between SLM and LM slightly, our analysis of the Rff data and vegetation cover still shows a significant impact of salvage-logging of living trees at SLM."*

It is not clear to me that there is a need for a conclusions section like the one you currently have. Possibly some comments on the limitations found and the implications of these results on wildfire consequences on carbon fluxes would be more interesting.

Biogeosciences requires a conclusions section therefore we will keep this section.

We have also added a new section on the limitations of the study to the discussion as stated in response to a previous comment.